# Analog content-addressable memories with memristors

Can Li [1✉], Catherine E. Graves [1✉], Xia Sheng[1], Darrin Miller [2], Martin Foltin[2], Giacomo Pedretti[1,3] & John Paul Strachan [1✉]

A content-addressable memory compares an input search word against all rows of stored words in an array in a highly parallel manner. While supplying a very powerful functionality for many applications in pattern matching and search, it suffers from large area, cost and power consumption, limiting its use. Past improvements have been realized by using memristors to replace the static random-access memory cell in conventional designs, but employ similar schemes based only on binary or ternary states for storage and search. We propose a new analog content-addressable memory concept and circuit to overcome these limitations by utilizing the analog conductance tunability of memristors. Our analog content-addressable memory stores data within the programmable conductance and can take as input either analog or digital search values. Experimental demonstrations, scaled simulations and analysis show that our analog content-addressable memory can reduce area and power consumption, which enables the acceleration of existing applications, but also new computing application areas.

[1] Hewlett Packard Labs, Hewlett Packard Enterprise, Palo Alto, CA 94304, USA. [2] Silicon Design Lab, Hewlett Packard Enterprise, Fort Collins, CO 80528, USA. [3] Dipartimento di Elettronica, Informazione e Bioingegneria, Politecnico di Milano, 20133 Milan, Italy. ✉email: can.li@hpe.com; catherine.graves@hpe.com; john-paul.strachan@hpe.com

To increase power efficiency and cost performance, there is growing interest in computing architectures that allow for in-memory processing[1] in order to reduce data movement and address the memory wall. In this vein, recent work has shown the promise of using non-volatile memory devices, or memristors, for accelerating matrix multiplication directly in memory arrays, accelerating a range of applications such as machine learning[2–6], analog signal processing[7,8], and scientific computing[9–11]. The performance improvements from this approach originate from two principles. First, computation is performed where the data are stored, removing the expensive power and latency costs of data movement between separate computing and memory units in a von-Neumann machine. Second, computation is performed in the analog domain, which provides exponential efficiency gains over digital, particularly at lower precision requirements. Each device performs analog computations that would otherwise require multiple digital elements. Despite the great promise of this approach, demonstrations have thus far been limited to the acceleration of matrix multiplication via crossbars.

Meanwhile, in-memory computational approaches in the digital domain have been extensively explored over the years[12]. While many proposed circuit typologies have not been implemented in commercial systems, content-addressable memory (CAM) and the related ternary CAM (TCAM) have stood as a notable exception[13,14]. CAM/TCAM circuits natively perform a matching operation between an input data word (search key) and a stored set of data patterns in the CAM/TCAM array. The operation is highly parallel and another example of an in-memory operation, leading to extremely high throughput compare operations at low latency, and therefore commercial success in applications such as network routing[15,16], real-time network traffic monitoring[17], and access control lists (ACLs)[18]. While powerful, CAM performance benefits come at the cost of large power and low memory density, limiting modern usage to high cost niche areas that demand high performance. Recent work has shown that utilizing non-volatile memristors (or resistive memory devices) in TCAM circuits reduces area and power[19–27] and provides the flexibility to accelerate powerful finite state machines, particularly for Regular Expression matching used in Network Intrusion Detection Systems[23,28]. However, nearly all memristor-based CAM designs utilize schemes similar to conventional static random-access memory (SRAM) designs where the memristor only encodes binary states. The highly tunable analog conductance in memristor devices, with many stable intermediate states are not leveraged[29]. An analog CAM design was proposed more than a decade ago that matches an input voltage with precise values stored in analog storage cells[30], but has not been implemented likely due to practical concerns of high power and area as it requires large numbers of active comparators and inefficient array implementations.

Here, we propose a memristor-based analog CAM that significantly increases data density and reduces operational energy and area for these in-memory processing circuits. Our analog CAM design stores a range of values in each cell using the tunable conductance of memristive devices, and compares an analog input with this stored range to determine a match or mismatch. The concept has been validated with proof-of-concept experiments, as well as simulations to establish performance and scalability. When used to store narrow ranges as discrete levels, our analog CAM can be a direct replacement of digital CAMs while providing higher memory densities and smaller power consumption. This may enable the use of CAMs for more generic scenarios[31–34] such as for associative computing that otherwise struggle with the limited memory densities and high power consumption of conventional CAMs. More importantly, our analog CAM can store wide intervals of continuous levels, thereby enabling novel search and matching functionality in the analog domain. The analog CAM cell presented here can also be searched with analog input signals, enabling the processing of analog sensor data without the need for an analog-to-digital conversion step.

## Results

### 6-transistors 2-memristors analog CAM

The proposed analog CAM concept is illustrated in Fig. 1, where analog voltage values are input to the analog CAM to be searched against the analog ranges encoded by multilevel conductances in the memristors. This is distinct from all previously reported CAMs (SRAM or memristor-based), where only digital signals are searched and stored (Fig. 1a). Similar to a digital CAM, the "match" signal for each row is

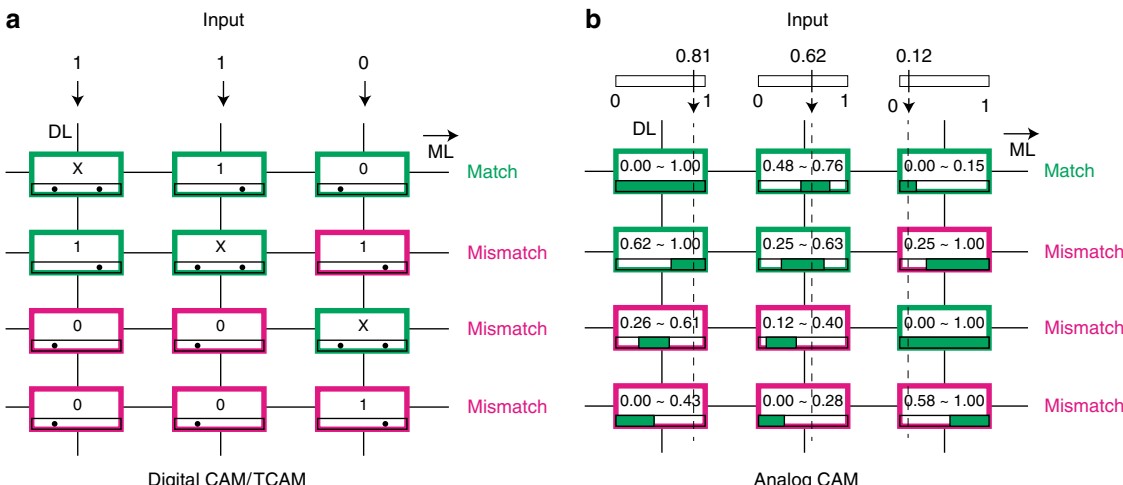

**Fig. 1 Schematic of the memristor analog content-addressable memory concept. a** A digital content-addressable memory (CAM) compares the input word against all stored words or rows in parallel. Ternary CAM (TCAM) is an extension where in addition to search/stored "0" and "1" values, "X" is a wildcard that always yields a match. Data are searched along vertical Data Lines (DL) and the binary match result of the compare operation between searched and stored words in each row is sensed on horizontal Match Lines (ML). The CAM returns the match location of stored data and the searched input (first row here). **b** The analog CAM searches and stores analog data, where the input data can be a continuous value, and the stored data are a continuous interval with a lower and upper bound representing an acceptance range for a match.

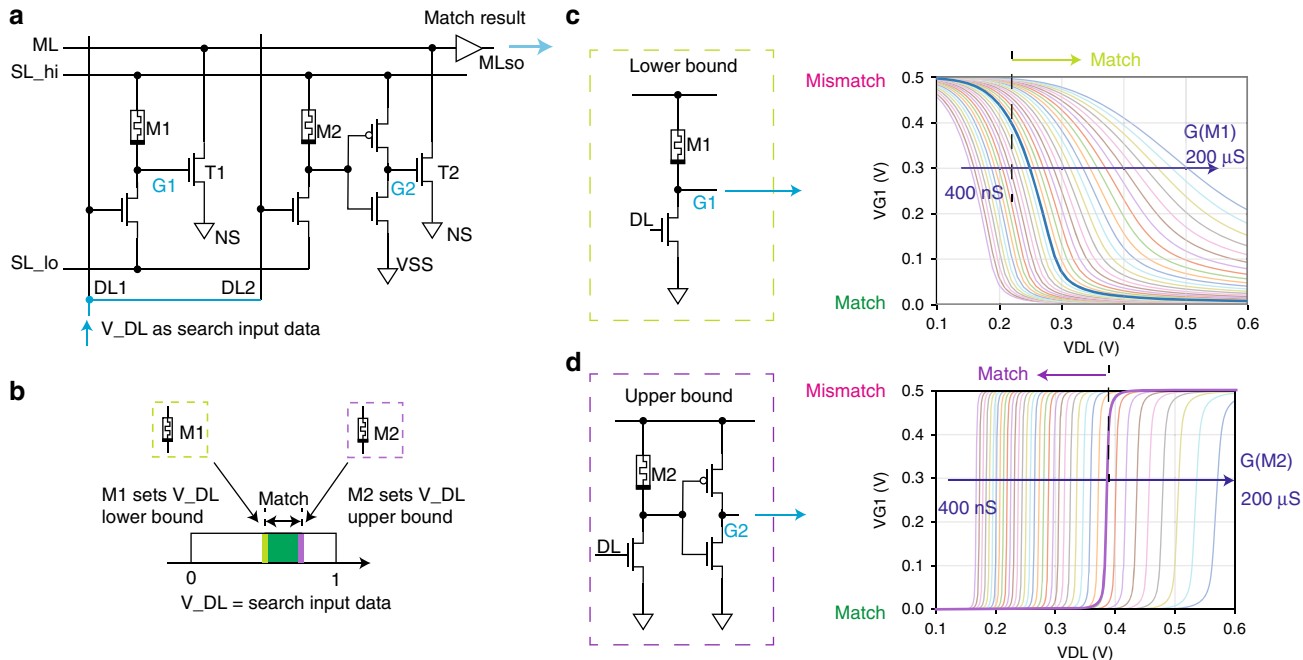

**Fig. 2 6-transistors 2-memristors analog content-addressable memory circuit. a** Schematic of our proposed analog CAM circuit, composed of six transistors and two memristors (6T2M). Voltage amplitude on the Data Line (DL) provides search input and the matching result is sensed as the voltage level on the Match Line (ML). **b** "Match" result when the analog input is within the range (narrow green band) stored by the cell. The stored range is defined by the conductances of two memristors (M1 and M2) in the cell, with M1 determining the lower bound and M2 determining the upper bound of the matching range. **c,d** Voltage divider sub-circuits translate the input voltage (a search value) to the gate voltage on the ML pull-down transistors. **c** When the input voltage is smaller than the lower bound threshold, the voltage on the gate of the T1 is large enough to pull down the ML, yielding a "mismatch" result. The lower bound threshold is set by the M1 memristor conductance. **d** Similarly, when the input voltage is larger than the upper bound threshold, which is tuned by the M2 memristor conductance, the cell returns a "mismatch" result by pulling down the ML. Here, SL_hi is at 0.5 V which sets the max G1 and G2 voltage.

generated on the matchline (ML) only when all the inputs for every column match the data stored in that row's memory. In contrast to digital CAMs, each analog CAM cell can match a range of analog input voltages (Fig. 1b), instead of a digital value. The analog CAM can be configured to match a narrow range of discrete values, and therefore one analog CAM cell is a direct functional replacement for multiple digital CAM cells. In addition, similar to storing a "wild card" or "X" in the TCAM, the proposed analog CAM can also store a range of continuous values, which would otherwise be difficult to implement with digital CAMs/TCAMs, but (as described later) is beneficial in internet packet (IP) routing, and more novel applications in decision trees, associative computing[34,35] and probabilistic computing.

To realize the proposed analog CAM concept, we have designed an analog CAM cell circuit where each cell is composed of six transistors and two memristors (6T2M) (Fig. 2a). The analog input search data are mapped to voltage amplitudes $V_{DL}$ applied along datalines (DL), and the stored analog range is configured by the programmed conductances of the two memristors of the cell (Fig. 2b). Similar to existing CAM circuit implementations, the search operation starts by precharging each row's ML to a high logic level, and the MLs stay high (match) only when all of the attached CAM cells of a row match the corresponding input, otherwise discharging and leading to a low logic level (mismatch) on the ML. In the 6T2M design, the ML is connected to pull-down transistors (T1, T2), and is kept high for a "match" result when the gate voltage of the pull-down transistors is smaller than the threshold voltage, keeping the transistor channel in a high resistance state.

Each analog CAM cell stores an upper and lower bound for matching against the input search value. These bounds are

encoded by two voltage divider sub-circuits which determine the gate voltages of two pull-down transistors connected to the ML. As shown in Fig. 2c, the voltage divider sub-circuit consists of a transistor and series-connected memristor, which generates the gate voltage (G1) of the pull-down transistor (T1) in the 6T2M analog CAM circuit to embody the analog CAM cell's lower bound match threshold. When $V_{DL}$ is larger than a certain threshold voltage, the transistor is highly conductive and thus the search voltage between SL_hi and SL_lo (typically at GND) will mainly drop across the M1 memristor, resulting in a small voltage on G1 that does not turn on the pull-down transistor and yields a match result. The lower bound of the input voltage $V_{DL}$ that yields a match is configured by tuning the memristor conductance in the voltage divider. The upper bound of the search range is configured similarly with an independent voltage divider using M2 and an inverter to control the gate voltage (G2) of the second pull-down transistor (T2) (Fig. 2d). This concept is shown by the simulation of the voltage on G1 and G2 depending on $V_{DL}$ with different M1 and M2 memristor conductances (Fig. 2c, d). As a result, the cell keeps ML high only when $V_{DL}$ is within a certain range as defined by the M1 and M2 conductances. As several cells are connected on the same ML in a row, just as in digital CAMs, a row ML outputs "high" only when each cell in the row matches.

**Simulations and experiments**. To validate our circuit design and further investigate the memristor-based analog CAM concept, we (1) conducted extensive simulations of individual analog CAM cells and CAM arrays and (2) experimentally measured analog CAM circuit operation in a taped-out silicon test chip. The circuit simulations shown here utilize 16nm design rules to enable

projected performance comparisons against current CMOS-based solutions, and our silicon tape-out utilized a 180nm technology node to provide voltage and current overheads and accelerate design to fabrication time.

We first validated the operation of the individual memristor analog CAM cell circuit with circuit simulations (see Methods for details) based on a layout using commercial 16 nm design rules (see Fig. 3a). The memristor conductance tuning in an analog CAM array is similar to the "write" operation in a 1T1M array and described in Supplementary Note 1, and Supplementary Fig. 1, with the parameters summarized in Supplementary Table 1. The current design prioritizes feasibility and demonstration of this new circuit concept and is not fully optimized for speed or power consumption.

The analog stored value was first configured in the CAM cell by setting the conductance of two memristors in one analog CAM cell to 40 µS and 80 µS and tested by applying different $V_{DL}$ values to observe the changing $V_{ML}$ behavior during the search operation. From simulations we see that after the search is initiated (by pulling SL_hi high), $V_{ML}$ stays high (Fig. 3b) when $V_{DL}$ is 0.4 V, indicating a "match", but is discharged low when $V_{DL}$ is either 0.3 V or 0.5 V for a "mismatch". The operation's timing diagram and voltage parameters are presented in Supplementary Fig. 2, from which one sees that the search result can be measured from the transient $V_{ML}$ at some time (e.g. 100 ps) following the search when the voltage difference (i.e. sensing margin) between match and mismatch scenarios is large enough for a sense circuit. The simulated ML sensing output at 100 ps following the search operation for different $V_{DL}$ (Fig. 3c) shows that this programmed memristor configuration

corresponds to matching for 0.37 V < $V_{DL}$ < 0.42 V. The lower bound (V_lo) and the upper bound (V_hi) of the analog CAM cell's acceptable matching range can be configured independently by tuning the corresponding memristor conductance in the cell. Using the resulting mapping between the voltage bound and conductances (Fig. 3d) as a guide, we configured analog CAM cells to match various voltage ranges (Fig. 3e) or eight narrower ranges (Fig. 3f) for representing 3-bit discrete voltage levels. CMOS process variation effects are studied with a layout-based simulation under different corner conditions, and results (in Supplementary Fig. 3) show that while different conditions slightly change the latency and search boundaries, we can still perform a calibration under those specific conditions to achieve the same bit accuracy as our search boundary is programmed with an iterative program-and-verify approach. Therefore, the proposed analog CAM cell implements the desired functionality and can be used to search for discrete levels (encoding at least 3-bits in a single cell) or for arbitrary analog voltage ranges to encode continuous values.

Next, we experimentally verified the proposed analog CAM operation in 6T2M analog CAM cells designed and fabricated at a 180 nm technology node on a silicon test chip (see Methods). Memristors of size 50 nm × 50 nm and based on Ta/TaO$_x$ were monolithically integrated in a Back End of the Line (BEOL) process with CMOS circuits on top of metal 6 and tungsten vias (Fig. 4a). Figure 4b shows the top view image of the analog CAM array with integrated memristors. The integrated memristors have a wide ~$10^3$ range of conductance tunability (Fig. 4c) and a programming voltage <1 V under direct-current (DC) sweeps (Fig. 4d shows a typical switching curves, and Supplementary

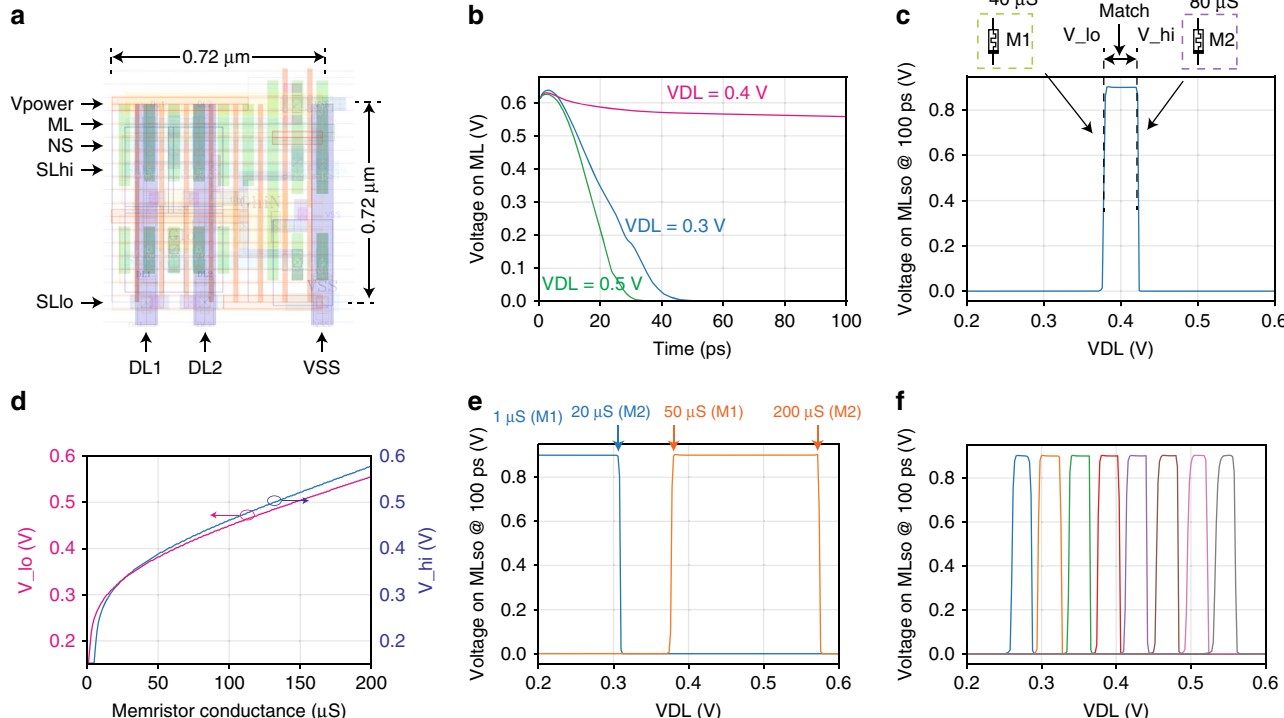

**Fig. 3 Simulations of the memristor analog content-addressable memory cell. a** The layout design of one analog content-addressable memory (CAM) cell in an analog CAM array using commercial 16 nm design rules. The transistors in this proof-of-concept design are over-sized to allow large driving currents. **b** Simulated transient voltage response on the Match Line (ML) where the different curves show searches with different Data Line (DL) voltages for a matching case (red) and two mismatching cases (blue, green). **c** The circuit simulation with the same memristor configuration shown in (**b**) shows that the cell matches a range of DL voltage, whose bounds are independently controlled by the conductances of the two memristors in the cell. **d** The simulated relation between the search range and the memristor conductance. The blue curve shows the lower bound of the stored range, while magenta shows the upper bound. **e, f** Using differently configured memristor conductances, the cells in the array can store (**e**) a continuous range of values, or (**f**) discrete levels (showing eight levels or 3-bits).

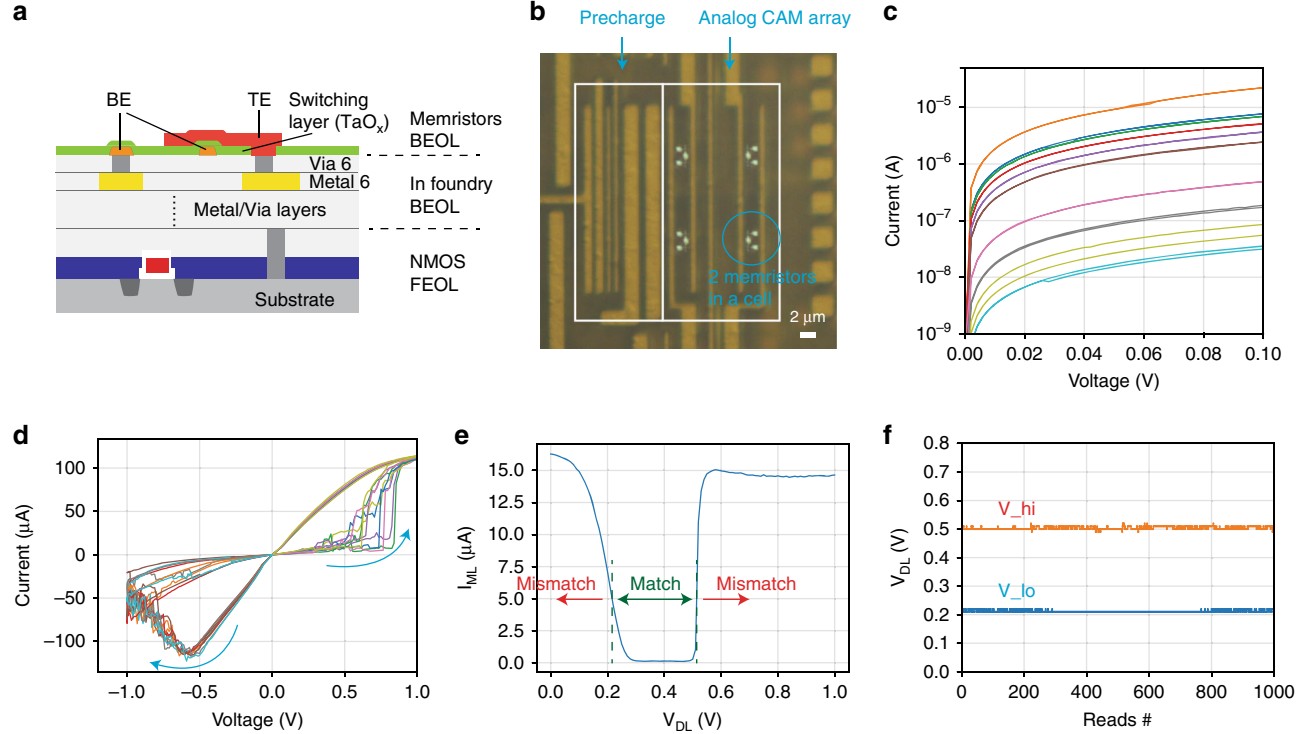

**Fig. 4 Experimental demonstration. a** Cross-sectional diagram showing monolithic integration of memristor on top of CMOS circuits. **b** An optical microscope image of an integrated analog CAM array with an on-chip precharging peripheral circuit. **c** Current–voltage relation of the memristor in the analog CAM circuit for many programmed states, showing the wide continuously tunable conductance range. **d** The conductance can be programmed from a low conductance state to a high conductance state by a large voltage (>0.5 V) on SL_hi, and reversed by a large voltage on SL_lo. **e** The Match Line (ML) pull-down current with respect to the applied data line voltage, after programming the memristors. **f** The analog CAM cell stored range was extracted from the pull-down current in Supplementary Fig. 6 (and one trace in (**e**)) where a high pull-down current (> 5 μA) indicates a mismatch. The range stayed unchanged for 1000 searches over more than 8000 s.

Fig. 4 shows a test chip under measurement), enabling us to validate the search operations experimentally. Given that the switching voltage of the memristor device exhibits a certain degree of variation, some devices may require a larger voltage to program than others (see Supplementary Fig. 5a). Therefore, it may impose challenges for a future technology node which supplies a voltage smaller than 1 V. We programmed two analog cells in the same row to store different ranges using an iterative program-and-verify approach and observed the match line (ML) pull-down current as we sweep the corresponding data line (DL), with the experimental configuration schematic shown in Supplementary Fig. 6a. As expected, the pull down current is low only when the applied data line voltage falls into the programmed range, indicating a match (Fig. 4e), and the other cell attached to the same ML shows a different search range (Supplementary Fig. 6b, c). Figure 4f shows 1000 repeated measurements without any observed disturb effects in the stored range (see also Supplementary Fig. 6d) and Supplementary Fig. 7 and 5 show the stability statistics of the analog conductance states of the memristors. These results suggest our analog CAM does not require static power to store a range table, nor frequent updates once programmed.

The relationship between an analog CAM cell's stored range for a match and the programmed memristor conductances can be understood by the series-connected transistor and memristor voltage divider (see Fig. 2c, d). During a search operation, the serial transistors in the divider are working in the triode regime, as the voltage drop across the transistor channel is fairly small. Under this condition, $V_{ML}$ stays high when $V_{DL}$ follows Eq. (1), with bounds from the lower bound M1 voltage divider and the

higher bound M2 voltage divider:

$$
\begin{aligned}
G_{M1} \cdot (V_{SLhi}/V_{TH,ML} - 1)/\beta + V_{TH} \\
\leq V_{DL} \leq G_{M2} \cdot (V_{SLhi}/V_{TH,inv} - 1)/\beta + V_{TH},
\end{aligned}
\tag{1}
$$

where $V_{TH}$, $V_{TH,ML}$, $V_{TH,inv}$ are the threshold voltages of the transistor in the M1 voltage divider, the T1 pull-down transistor, and the inverter respectively. $\beta$ ( $= \partial G_T/\partial V_{DL}$) is a constant coefficient in the transistor transfer function. $G_{M1}$ and $G_{M2}$ are the memristor conductances, which are linearly related to the accepted $V_{DL}$ for a match according to the above equation. This analysis is consistent with the results shown in Fig. 3d when the transistor is working in the linear regime, but when the memristor conductance is very small (i.e. the transistor voltage drop is small) a numerical simulation is required to extract the precise relation. In practice, we use the simulated relation in Fig. 3d to determine the programming target memristor conductance values from the desired stored analog value or range. Under this assumption, we scale up our simulations from single cells to large analog CAM arrays to predict performance.

**Memristor analog content-addressable memory arrays.** While we have demonstrated single analog CAM cell operation in a small array, it is crucial to investigate whether large arrays can be operated without degradation to the desired search operation result. Using extracted parasitic parameters from the 16 nm layout, we constructed analog CAM arrays with arbitrary numbers of rows and columns (see Methods) to study how the analog CAM performs with increasing array size. Fig. 5a shows the simulation configuration, where the two memristors in each of

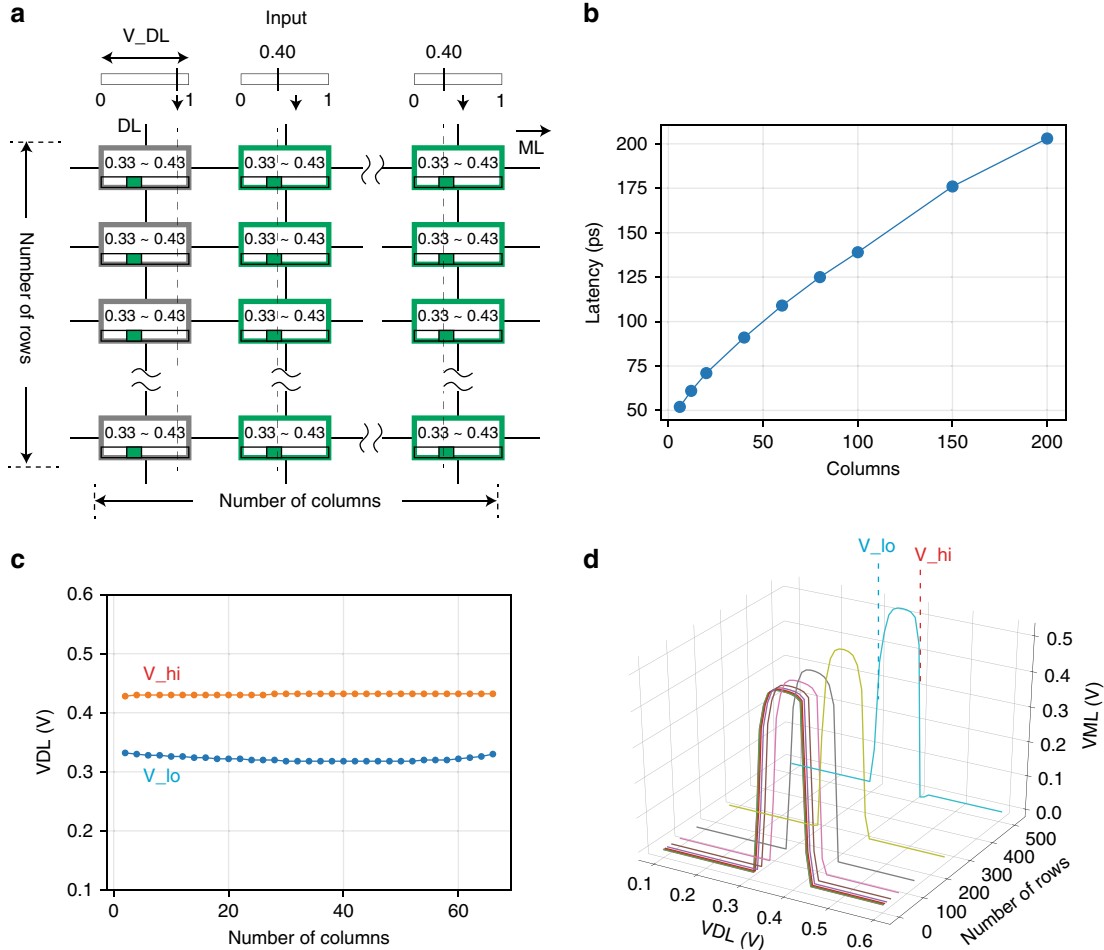

**Fig. 5 Analog content-addressable memory arrays. a** Simulated analog content-addressable memory (CAM) arrays with different sizes using extracted parasitic parameters. **b** The search latency in the one-bit mismatch case increases with the number of columns due to the increased ML capacitance. **c** There is a noticeable change in the acceptable search range with an increasing number of columns. In the worst case, up to 20 mV change is observed in an array of 64 columns. This still allows the capability to store and search 4-bits of information per analog cell across the 64 columns, but going beyond this would be challenging with the present circuit. **d** On the other hand, the change with increasing number of rows is negligible up to 512 rows. The three-dimensional (3-D) plot shows the sensed ML voltage with respect to the input voltage applied to the DL in arrays with an increasing number of rows (or words). High logic level on the ML indicates that the searching input matches the stored memory, where the lower and the higher bounds ($V\_lo$ and $V\_high$) are labeled in the plot. With an increasing number of rows the searching range of one cell in the array stays stable.

the analog CAM cells are programmed to 20 and 80 μS in order to accept $V_{DL}$ from 0.33 to 0.43 V. All DLs are biased to 0.4 V, except for one column DL that is swept from 0.0 V to 1.0 V to observe how $V_{ML}$ changes. This single-bit mismatch is the worst-case scenario as it represents the situation where the mismatch $V_{ML}$ decay is the closest to the match $V_{ML}$ behavior. Since all cells with $V_{DL} = 0.4$ V match, the $V_{ML}$ drop leading to a "mismatch" is initiated by the cells in the column with the sweeping DL. Similar to a conventional CAM, ML discharging latency increases with the number of columns because of larger ML capacitances (Fig. 5b). With increasing number of rows, on the other hand, the latency only changes 5% in our simulation for an array with 512 rows, suggesting our analog CAM supports the search for many entries in parallel.

As expected even with conventional CAMs, increasing the number of columns can also lead to a degradation of the $V_{ML}$ (Fig. 5c) such that the acceptable search range is slightly changed. Our analysis shows that the degradation of $V_{ML}$ with increasing column number is from the sub-threshold leakage current of the pull-down transistors (see Supplementary Note 2 for more details), and it can be improved with some emerging devices with extremely low leakage current and sharp transition, as we preliminary

explored in Supplementary Note 3, Supplementary Fig. 8, Supplementary Fig. 9, and Supplementary Fig. 10. Nevertheless, the change in the accepted voltage range is within 0.02 V and is thus sufficient to separate more than 20 discrete levels between 0.2 V and 0.6 V for 4 bit searching capability in a 64-column analog CAM array. On the other hand, analog CAM arrays with two columns but an increasing number of rows show little change (Fig. 5d) in $V_{ML}$ with additional rows (simulated up to 512 rows), demonstrating negligible row-wise interference.

**Applications**. The direct advantages of our proposed analog CAM are the improvements in energy and area over existing digital approaches. To demonstrate the potential scale of these improvements, we compared our analog CAM approach with the digital-equivalent for the usecase of classifying Internet protocol (IP) packets, which is a common commercial application for CAMs[13]. The ternary wildcard "X" capability of TCAMs is frequently used to compress multiple table entries into fewer rows in the IP routing look-up table, owing to the fact that most classifying ranges are continuous. With our proposed analog CAM's ability to store broad ranges, this look-up table can be further

compressed. Analysis in the previous section suggested that one analog CAM cell is capable of searching 8-64 discrete levels, depending on the size of array and the specific implementation. Columns can therefore be combined as fewer cells are required to store the same amount of information. Additionally, by taking advantage of the range storage capability, fewer rows are required than in a digital CAM/TCAM representation. A real example is given in Supplementary Note 4 and Supplementary Fig. 11, which shows a 14× reduction in the number of required cells from conventional TCAM to 16-level analog CAM cells, with further reductions possible with improved analog CAM cells. As an area improvement, given the analog CAM cell occupies less area than the SRAM cell, we estimate an 18.8× reduction for our analog CAM table with 12.5 μm² chip area compared to an SRAM implementation with 235.2 μm² (see Supplementary Section 4 for details).

To evaluate the search energy improvement with an analog CAM, we simulated the circuit current from all the power supplies with an 86×12 analog CAM array. For a practical evaluation of all digital applications, we custom designed a digital-to-analog (DAC) converter, which imposes additional overhead in both chip area and energy, but our analyses in Supplementary Note 5, Supplementary Fig. 12 and Supplementary Table 2 show this overhead is limited to approximately 10%. The cumulative consumed energy is calculated by integrating the voltage and current over the 16-cycle search with all MLs discharging in the search for the worst-case scenario. Estimating the full array power (including drivers and unoptimized peripherals such as the custom-designed digital-to-analog (DAC) converters), the average total energy per search is ~0.52 fJ per analog cell, or 0.037 fJ for the equivalent number of TCAM bits implementing the same function (see Supplementary Note 4 for details). The energy per cell consumption is significantly smaller than an SRAM TCAM (0.165 fJ)[36], which utilizes numerous power saving techniques that provides a > 10× reduction but are not implemented in our analog CAM yet, and a conventional memristor TCAM (0.17 fJ)[37].

In addition to serving as a higher data-density digital replacement, the proposed analog CAM offers novel applications when the range search capability is utilized. As an example, a decision tree with binary and non-binary classification features can be implemented in the analog CAM directly by mapping each root to leaf path to a row in the analog CAM (see Supplementary Fig. 13 and Supplementary Note 6 for mapping details). Logically, each root-to-leaf path traverses a series of nodes with Boolean ANDs between elements in a given input feature vector (Supplementary Fig. 13a). Since AND is commutative, we can reorder the nodes such that feature variables are processed in the same order for all paths[38]. Nodes for the same feature are combined into one node and "don't care" nodes can be inserted for features absent from a specific path, such that each path is of equal length. This representation can then be directly mapped to the analog CAM array, with each root to leaf path a row. As the matching row can directly drive the readout of the classification result, tree traversal becomes a one-cycle operation (Supplementary Fig. 13b). This high throughput and low latency operation is highly advantageous and differentiated from current usage. While ensemble tree-based models are a popular state-of-the-art machine learning approach for classification and regression across diverse real-world applications, these models are difficult to optimize for fast runtime without accuracy loss in standard architectures[32] due to non-uniform memory access patterns, resulting in unpredictable traversal and classification times today. With our proposed analog CAM, it becomes feasible to process large tree-based models at high data rates, such as those required for streaming applications or autonomous vehicles.

**Discussion**. In summary, we have proposed an analog CAM cell circuit taking advantage of the analog memristor conductance tunability for the first time. A practical circuit implementation composed of six transistors and two memristors has been demonstrated in both experiment and simulation. The analog CAM increases memory density significantly, as one analog CAM cell can store multiple bits with only six transistors while an SRAM CAM cell stores 1 bit values with 10 transistors, or ternary values with 16 transistors in a TCAM cell. The analog capability opens up the possibility for directly processing analog signals acquired from sensors, and is particularly attractive for Internet of Things applications due to the potential low power and footprint. The output of the analog CAM after the sense amplifiers is digital, and thus can also remove the analog-digital conversion cost entirely. Finally, the functionality of our analog CAM with interval storage is intrinsically different from digital CAMs, which may enable new computing applications in decision tree models, associative computing and probabilistic processing where inexact compares and real-valued analog transition probabilities are common.

## Methods

**Memristor integration**. The memristors are monolithically integrated on CMOS fabricated in a commercial foundry in a 180 nm technology node. The integration starts with a removal of silicon nitride and oxide passivation with reactive ion etching (RIE) and a buffered oxide etch (BOE) dip. Chromium and platinum bottom electrodes are then patterned with e-beam lithography and metal lift-off process, followed by reactive sputtered 4.5 nm tantalum oxide as switching layer. The device stack is finalized by e-beam lithography patterning of sputtered tantalum and platinum metal as top electrodes.

**Circuit simulation for analog CAM cell and arrays**. The proposed 6T2M analog CAM cells designed in the Cadence Virtuoso Custom IC design environment (version ICADV12.1-64b.500.14), and the simulation result is analyzed and post-processed with HP-SPICE (version 4.11). The simulations utilize the TSMC 16 nm library and the designs follow the corresponding rules. The voltage parameters and timing diagram are shown in Supplementary Fig. 2. A custom python script generates the netlist for analog CAM arrays with different numbers of rows and columns and arbitrary configured memristor conductances and input voltages. In the netlist, the parasitic parameters are extracted from the taped out layout, including the wire resistance 1.91, 2.27, 0.85 Ω per block for ML, DL, SL, and capacitance 0.227 fF, 0.324 fF, 0.454 fF between different analog CAM cells. The voltage stimulus is always applied to the nodes that are the furthest from the ML sensing node, so that the impact of the wire resistance is the most significant, *i.e.* the worst case scenario.

**Electrical characterization**. The electrical characterization is conducted with a semiconductor parameter analyzer (Keysight B1500A) and Cascade probe station under room temperature. The conductance programming is performed with quasi-static direct-current (DC) sweeps, and where the current through the device is limited and controlled by the series-connected transistor. After programming, the memristor conductance is readout by applying a small voltage across the memristor with the series-connected transistor fully turned on. The search operation is conducted by applying and sensing voltages from the corresponding node to/from a source measurement unit (SMU) on the B1500A. A custom python code with PyVISA library was used to control the equipment.

## Data availability
The data supporting plots within this paper and other findings of this study are available with reasonable requests made to the corresponding author.

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

## Acknowledgements

The authors acknowledge fruitful discussions with Jim Ignowski and Yuanming Zhu. This research was based upon work supported by the Office of the Director of National Intelligence (ODNI), Intelligence Advanced Research Projects Activity (IARPA), via contract number 2017-17013000002, and by the Army Research Office and was accomplished under Grant Number W911NF-19-1-0494. The views and conclusions contained in this document are those of the authors and should not be interpreted as representing the official policies, either expressed or implied, of the ODNI, IARPA, Army Research Office or the U.S. Government. The U.S. Government is authorized to repro-duce and distribute reprints for Government purposes notwithstanding any copyright notation herein.

## Author contributions

C.L., C.G., J.P.S. contributed to the conception of the idea and wrote the manuscript. C.L. designed the experiment and collected the simulation and experimental data. X.S. inte-grated the memristors, D.M. did the layout design, M.F. and C.L. designed the custom digital-to-analog converter. C.G., C.L., G.P., J.P.S. contributed to the idea of mapping the decision tree.

## Competing interests

The authors declare no competing interests.
