## [Peer Review File · Nature Communications]

Reviewers' comments:

Reviewer #1 (Remarks to the Author):

- In this work, Li and coauthors propose an analog content access memory, using memristors. The paper is well written, and this is actually a neat idea. But to a large extent, the paper confused me, and it is difficult to conclude if the concept is valuable based on the results presented in the paper.

- This is not an experimental paper. The experiment is only a proof of concept with a large area device used as a programmable resistor. There are only a few measurements, and they only show the functionality, which is not surprising at all because the concept in itself is not complicated. What I would have been interested to see from experiments is maybe statistical measurements, endurance measurements, comparison of programming conditions wrt. CAM functionality, speed analysis, etc. But this is not present.

- On the other hand the paper presents circuits simulations. These simulations are surprisingly based on a 180 nm technology, which is very old and quite different from modern CMOS, which is much lower voltage, and has very different leakage and variability. There are all important questions for the CAM design:

1. Because modern CMOS is low voltage, and memristor require programming voltage $> 1V$, you will probably need to use high voltage transistors. Due to this the area benefit wrt. SRAM (which uses small size transistors) will not be as high as claimed here.

2. Leakage is a very important question for scalability of the design (as shown by the authors), so the maximum CAM size might be smaller than stated here in modern CMOS.

3. CMOS variability is ignored, although it seems a very important question. CMOS leakage is very much affected by variability, and only one particularly leaky CMOS transistor could kill the circuit.

- Overall, I found that the circuit simulations were not so interesting. There is no discussion on how to size the elements of the circuit to optimize speed / energy / robustness / scalability.

The authors clearly state: "The current design prioritizes feasibility and demonstration of this new circuit concept, and is not yet optimized for speed or power consumption" . But what is the point, as the circuit has not been demonstrated anyway? I would really have done the study with scaled CMOS.

- The part about the volatile memristor/emerging threshold switching memristor ignores the question of reliability. My understanding is that the volatile memristor might age very quickly as search operation will switch it. I feel that this question should be included in the study.

- The energy analysis is very qualitative : "We also expect a similar reduction in operational power with the analog CAM cell in comparison to conventional TCAMs, as a major portion of the dynamic energy consumption for a CAM operation is charging parasitic wire capacitances, and the reduced cell count and area also results in shorter wires and reduces total wire capacitance".

I would have expected detailed comparison with digital memristor-based TCAM, and also SRAM-based TCAM.

This qualitative point becomes an assertive: "analysis showing that our analog CAM can reduce area and power consumption (37×) compared to a digital version." in the abstract.

Reviewer #2 (Remarks to the Author):

This work describes the concept of analog CAM and implements the proof-of-concept demonstration. The 6T2M analog CMA cell is proposed to realize the function, and the analog CAM cell takes advantages of multi-level memristor conductance to represent different matched ranges. In this manner, it stores multi-bits or a continuous range of content in CAM applications. The idea is

interesting. However, the major results are shown by simulations which limits the impact of this work. More experimental data are needed to validate the application of analog CAM. Moreover, the benefits from the using of analog CAM and the comparison to conventional CAM should be carefully discussed. The detail comments are listed below:

1. A potential important contribution of this manuscript is the experimental demonstration of the proposed 6T2M analog CAM cell. However, the exiting manuscript has some misleading description of the experimental part. Based on this reviewer's understanding, the authors only measured 1T1R devices. Did the authors physically constructed a 6T2M cell and measure the electrical data from the cell? If this is the case, the authors should clearly show how to integrate the experimental part and simulation part in the man text with figures. Also, this reviewer strongly suggests the authors should conduct the experiment with physical 6T2M cell.

2. As the author declared, novel memristor devices are suitable for analog in-memory-computing at low precision requirements. However, the application of analog CAM requires reliable and accurate interval states of memristor devices. However, the memristor device suffers from some non-ideal characteristics, such as device variances, conductance drift and so on. Please show more data about the reliability behavior of the proposed analog CAM. Besides, according to Fig. 4e, the experimental output exhibits significant distortion. Please explain and comment.

3. The performance comparison of analog CAM to conventional CAM is not fairly evaluated. When benchmarking the performance of the analog CAM, the cost of the necessary periphery blocks, e.g. the DACs, circuits to program and verify the cell conductance range, should be included and carefully discussed.

4. The manuscript takes non-ideal factors of transistors into account, such as sub-threshold leakage current, etc., but memristors also have non-ideal factors, such as device read noise, retention, etc. The authors should discuss those non-ideal factors impact on the analog CAM search results, in other words, on the upper and lower bounds.

5. Similar to comment 1, how will the practical imperfections of the volatile memristors affect the system performance in the 4T2M2S analog CAM model, Figure 6? The simulation is the ideal case and the authors should include the discussion of non-ideal factors. Meanwhile, is it possible to realize an inverter with 0.4V logic output under 180nm rule (i.e. the MLso)? Please explain.

6. Could the author explain more about how the non-linear effects are introduced when deducing the analytic expression of VDL in page 12?

7. This article introduces the circuit design of analog CAM based on non-volatile memory, and compares its advantages with the traditional digital CAM. However, there has been analog CAM design before. What is the advantage of this article over previous design (Analog content addressable memory (CAM) employing Analog nonvolatile storage, US patent 6985372, 2006)?

Reviewer #3 (Remarks to the Author):

This paper proposes the nonvolatile analog TCAM with 6T2M or 4T4M schemes. The idea is novel and interesting, however, more results are needed to demonstrate the novelty and potential of this work. Therefore, the authors should revise their manuscript to address the concerns below before a final decision is reached.

1. The authors should show 6T2M/4T4M cell with feature size as the unit.
2. The authors give a clear view of the concepts and explanation of the aTCAM cell and implementation. However, the authors should use the cell to build more applications in addition to IP routing to differentiate this work from the existing nonvolatile TCAM, because the latter also improves the density. Perhaps, as the authors propose, the fuzzy logic or probabilistic processing.
3. For TCAM, nanosecond order of magnitude is too slow. Even this paper it is proof of the concept, the authors better show the potential way to boost the search speed.
4. The authors better add a diagram or cross-section microscopic picture to demonstrate the monolithic integration with silicon transistors.

We thank the reviewers for constructive comments and acknowledging the merit of the idea. We have revised the manuscript based on the feedback. Specifically, we have conducted further experiments on the physical 6T4M cell based on 180 nm CMOS after a recent tape-out and nanoscale memristor integration (new Figure 4), simulated the performance based on modern 16 nm CMOS (new Figure 3 and Figure 5) and custom designed peripheral (DAC) circuits (discussion on Page 17-18 and Supplementary Section 5 on Page 42-44), and added a proposed analog decision tree application to complement our direct digital replacement example on IP routing. These new contents address the novelty concerns and include both new measurements and new simulations. In the following point-by-point responses, the original reviewers' comments are in black fonts and our responses are in blue fonts. Changes in the revised main text and SI are highlighted with shaded background.

Reviewer #1 (Remarks to the Author):

1- In this work, Li and coauthors propose an analog content access memory, using memristors. The paper is well written, and this is actually a neat idea. But to a large extent, the paper confused me, and it is difficult to conclude if the concept is valuable based on the results presented in the paper.

RESPONSE: We would like to thank the reviewer for acknowledging the concept proposed in this manuscript is a neat idea. To validate our concept, we have conducted further experiments from the physical 6T4M analog CAM cell with 180 nm transistors and projected performance based on 16 nm layout-based simulation.

2- This is not an experimental paper. The experiment is only a proof of concept with a large area device used as a programmable resistor. There are only a few measurements, and they only show the functionality, which is not surprising at all because the concept in itself is not complicated. What I would have been interested to see from experiments is maybe statistical measurements, endurance measurements, comparison of programming conditions wrt. CAM functionality, speed analysis, etc. But this is not present.

RESPONSE: We agree with the reviewer that the present manuscript is not focused on statistical experimental measurements, as this work focuses on the new concept of an analog CAM cell. Largely because of the simplicity of the new concept design, we are confident of the feasibility, as well as based on extensive existing studies on memristive devices both from our group and many others. The most important property of the memristor with respect to the analog CAM implementation, in our view, is the stability and retention performance to store a range reliably within the designed cell. We have previously demonstrated memristors can be reliably programmed with high precision, and more importantly, they are able to maintain their states for a prolonged period of time (see Figure R1 adapted from [1]). **This feature has also been validated with our recent experiment (see Figure R2) in a physical 6T2M analog CAM, which shows the stored range did not change for 1,000 reads taking place in over 8,000 seconds.**

Figure R1. f, Histogram of the initial difference between the target and measured conductance written into a 128×64 array. A fit of the peak to a normal distribution yielded a standard deviation of $6 \mu\text{S}$, with the peak maximum located at $-5 \mu\text{S}$. g, Room-temperature state retention and read disturb of the device states. The d.c. conductance states of all the devices were measured with a 0.2 V bias for 1,000 cycles, or a total of 6.4 h, showing no discernible drift in the plots. h, Histogram of the normalized standard deviation (s.d.), defined as the s.d. per conductance range ($100\text{--}900 \mu\text{S}$), for all measured states in (g) fitted to a lognormal distribution. This shows that there are fluctuations during the read operation that can occasionally degrade the effective precision of an individual memristor, but 90% of the device states have a normalized s.d. less than 0.39% over 1,000 reads. Adapted from [1].

Figure R2. (left) The match line pull-down currents with respect to the input search voltage V_{DL} . The experiment is repeated 1,000 times in about 8,500 seconds, and no obvious change in the pull-down current is observed, indicating the search range can be reliably stored in the analog CAM without the need of static power for a prolonged period of time. (right) The searching range is extracted from the ML pull-down current during the 1,000 reads. **The figures have been added in the new text and supplementary information of the revised manuscript.**

The programming performance of the memristors, *i.e.* endurance and speed, is less critical, since the memristor conductances do not need to be frequently updated for most analog CAM applications, as long as they can be reliably programmed to their conductance states with iterative approaches. Previously we have reported they can be programmed with 5-6 bit precision [1], and the same/similar device has also demonstrated excellent endurance ($>10^{10}$) and programming speed ($<5 \text{ ns}$) in our team's prior studies.

Figure R3 The Ta/TaOx/Pt device can be switched for more than 10^{10} cycles. [2]

Figure R4 (c) The device can be repeatedly switched between HRS and LRS with 5ns pulses (SET: 2.2V; RESET: -4V) indicating faster than 5ns switching speed. [3]

We have revised the manuscript text to make the above points clearer.

[1] C. Li, et al, Analog signal and image processing with large memristor crossbars, Nature Electronics, 1, 52, 2018

[2] J. Yang, et al, High switching endurance in TaO_x memristive devices, Applied Physics Letters, 97, 232102, 2010

[3] H. Jiang, et al, Sub-10 nm Ta Channel Responsible for Superior Performance of a HfO₂ Memristor. Sci Rep 6, 28525 (2016).

3 - On the other hand the paper presents circuits simulation. These simulations are surprisingly based on a 180 nm technology, which is very old and quite different from modern CMOS, which is much lower voltage, and has very different leakage and variability. There are all important questions for the CAM design:

RESPONSE: The revised manuscript is now focused on 16 nm to project the performance with modern CMOS (Fig. 3&5 in the main text and Suppl. Fig. 2&3 in SI). The 180 nm simulations in the original manuscript was post-layout simulation to support a prototype tape-out and experiments.

3.1. Because modern CMOS is low voltage, and memristor require programming voltage $> 1V$,

you will probably need to use high voltage transistors. Due to this the area benefit wrt. SRAM (which uses small size transistors) will not be as high as claimed here.

RESPONSE: While some memristors require a programming voltage larger than 1 V, this is not universal. Our recent experiments have shown that our integrated nanoscale memristor can be switched reliably under 1 V of applied voltage under DC biases (see the following Figure R5 taken from the new Figure 4d in the revised manuscript). It may still be a valid challenge, particular for high speed programming, but it should not be a fatal challenge, as evidenced by many commercial sub-28 nm ReRAM based embedded memories.

Indeed, given the low switching voltage, we have also lowered the search voltage on SL_hi to 0.5 V (voltage drop on the memristor is always between -0.5 V and 0 V), so that the conductance state is not disturbed during the search operation. The low switching current has also enabled us to use transistors with smaller area in the new 16 nm simulation which we added in the revised manuscript. We have updated the area estimation in the revised manuscript based on our unoptimized layout, and the number is lower than our original estimation, but still noticeably advantageous considering our analog CAM can store multiple bits in a single cell.

Figure R5. The integrated nanoscale (50nm) memristor is repeatably switched between a low resistance state and a high resistance state for 10 cycles. (Taken from the new Fig. 4d in the revised manuscript)

Memristor device improvement is still under extensive research development. Some references e.g. [1] have shown a significantly smaller switching voltage along with other good performances, but there are still plenty rooms to improve to the device perspective. This work only proposes a concept that is feasibly with current technology, but at the same time it also provides a motivation for continuous device and material studies, in particular with respect to low voltage programming of analog conductance states.

[1] Y. Burgt, *et al*, “A non-volatile organic electrochemical device as a low-voltage artificial synapse for neuromorphic computing”, *Nature Materials*, 16, 414, 2017

3.2. Leakage is a very important question for scalability of the design (as shown by the authors), so the maximum CAM size might be smaller that stated here in modern CMOS.

RESPONSE: Following the reviewer’s suggestion, we have performed layout-based simulation with TSMC’s 16 nm design rule. As the reviewer suggested, the maximum CAM size is slightly

smaller than the original 180 nm simulation. We found the problem is primarily due to the non-zero wire resistance which caused inaccurate voltage reference and RC delays when the array grows. While many parameters may be tuned to optimize the performance in the future, the current design can still support an array with width of 64 and height of 512.

We have revised the following corresponding text (on Page 15 in the revised manuscript) to reflect the discussions above.

“Similar to a conventional CAM, ML discharging latency increases with the number of columns because of larger ML capacitances (FIGURE 5b). With increasing number of rows, on the other hand, the latency only changes 5% in our simulation for an array with 512 rows, suggesting our analog CAM supports the search for many entries in parallel.

As expected even with conventional CAMs, increasing the number of columns can also lead to a degradation of the V_{ML} (FIGURE 5c) such that the acceptable search range is slightly changed. Our analysis shows that the degradation of V_{ML} with increasing column number from the subthreshold leakage current of the pull-down transistors (see SUPPLEMENTARY SECTION 3 for more details), and it can be improved with some emerging devices with extremely low leakage current and sharp transition, as we preliminary explored in SUPPLEMENTARY SECTION 4. Nevertheless, the change in the accepted voltage range is within 0.02V and is thus sufficient to separate more than 20 discrete levels between 0.2 V and 0.6 V for 4 bit searching capability in a 64-column analog CAM array. On the other hand, analog CAM arrays with two columns but an increasing number of rows show little change (FIGURE 5d) in V_{ML} with additional rows (simulated up to 512 rows), demonstrating negligible row-wise interference.”

3.3. CMOS variability is ignored, although it seems a very important question. CMOS leakage is very much affected by variability, and only one particularly leaky CMOS transistor could kill the circuit.

RESPONSE: Following the reviewer’s suggestions, we studied the CMOS variability with post-layout simulation under different corner conditions such as process variation, etc. The result shows that while different conditions slightly change the latency number and the search boundaries, since our search boundary is programmed with an iterative program-and-verify approach, we can perform a calibration under that specific condition, like most analog circuits do. In case of a significantly leaky CMOS transistor, we should treat it as a defective device and disable the entire row or column. In the revised manuscript (in the revised text on Page 11 and Supplementary Figure 3), we have included the above discussion.

“CMOS process variation effects are studied with a layout-based simulation under different corner conditions, and results (in SUPPLEMENTARY FIGURE 3) show that while different conditions slightly change the latency and search boundaries, we can still perform a calibration under those specific conditions to achieve the same bit accuracy as our search boundary is programmed with an iterative program-and-verify approach.”

Supplementary Figure 3: Simulation of different process corners. Plots show the ML decay during a search in an 86×12 analog array with DL voltage of a, 0.3V (Mismatch), b, 0.4V (Match), c, 0.5V (Mismatch). The memristor are configured to the same conductance range with that in our single-device simulation in FIGURE 3b.

4- Overall, I found that the circuit simulations were not so interesting. There is no discussion on how to size the elements of the circuit to optimize speed / energy / robustness / scalability. The authors clearly state: "The current design prioritizes feasibility and demonstration of this new circuit concept and is not yet optimized for speed or power consumption". But what is the point, as the circuit has not been demonstrated anyway? I would really have done the study with scaled CMOS.

RESPONSE: The revised manuscript now includes studies on scaled 16 nm CMOS, which shows much better performance in terms of search latency and energy consumption. We also added some studies on latency dependence on array size and a more practical power consumption analysis.

The 16 nm circuit is still not fully optimized. There are many more parameters to explore for optimizing the performance, and in many cases, the optimization should aim at a particular application use case. Such work is a priority for future work, but we believe the currently proposed idea has been demonstrated with both simulation and experiment and demonstration of this new analog CAM concept is the core focus of this manuscript.

5- The part about the volatile memristor/emerging threshold switching memristor ignores the question of reliability. My understanding is that the volatile memristor might age very quickly as search operation will switch it. I feel that this question should be included in the study.

RESPONSE: The reviewer is right that it is possible that a diffusive (Ag-based) memristor may age during the search operation (the cite reference still showed $>10^8$ endurance cycles, see below), but devices with other mechanisms, such as tunneling, or IMT, are not likely to suffer the same problem. The goal of this part of the work is to provide a possible improvement direction for future studies, and at the same time give a motivation to develop a device with a specific property. To avoid distraction, we have moved this particular part to the supplementary information in the revised manuscript.

Figure R6, (b) Endurance of the selector with over 108 cycles. The inset shows the waveform employed in the endurance measurement which consists of a 10 μs pulse with an amplitude of 4 V for ON switching followed by a 0.1 V read pulse. The time interval between switching pulses is 100 μs (25 μs waiting time plus 75 μs read time). A series resistor of 3.9 M Ω was used to limit the ON-state current. Taken from [1]

[1] R. Midya, et al. “Anatomy of Ag/Hafnia-based selectors with 10^{10} nonlinearity”. *Advanced Materials*, 29, 1604457 (2017).

6- The energy analysis is very qualitative : “We also expect a similar reduction in operational power with the analog CAM cell in comparison to conventional TCAMs, as a major portion of the dynamic energy consumption for a CAM operation is charging parasitic wire capacitances, and the reduced cell count and area also results in shorter wires and reduces total wire capacitance”.

I would have expected detailed comparison with digital memristor-based TCAM, and also SRAM-based TCAM.

This qualitative point becomes an assertive: “analysis showing that our analog CAM can reduce area and power consumption ($37\times$) compared to a digital version.” In the abstract.

RESPONSE: We would like to thank the reviewer for the constructive comment. The revised manuscript has now included a more practical comparison based on 16 nm design with peripheral DAC taken into consideration.

The new discussion has been added in the revised manuscript as follows:

“Additionally, by taking advantage of the range storage capability, fewer rows are required than in a digital CAM/TCAM representation. A real example is given in SUPPLEMENTARY SECTION 5, which shows a $14\times$ reduction in number of required cells from conventional TCAM to 16-level analog CAM cells, with further reductions possible with improved analog CAM cells. As an area improvement, we estimate an $18.8\times$ reduction for our analog CAM $12.5 \mu\text{m}^2$ table compared to an SRAM implementation $235.2 \mu\text{m}^2$ (see SUPPLEMENTARY SECTION 5 for details).

To evaluate the search energy improvement with an analog CAM, we simulated the circuit current from all the power supplies with an 86×12 analog CAM array. The cumulative consumed energy is calculated by integrating the voltage and current over the 16-cycle search with all MLs discharging in the search for the worst-case scenario. Estimating the full array power (including drivers and unoptimized peripherals such as the digital-to-analog (DAC) converters), the average total energy per search is ~ 0.52 fJ per analog cell, or 0.037 fJ for the equivalent number of TCAM bits implementing the same function (see SUPPLEMENTARY SECTION 5 for details). The energy per cell consumption is significantly smaller than an SRAM TCAM

(0.165 fJ)³⁶, which utilizes numerous power saving techniques that provides a >10× reduction but are not implemented in our analog CAM yet, and a conventional memristor TCAM (0.17 fJ)³⁷.”

With details described in the Supplementary Information:

“To evaluate the search energy consumption, our analog CAM array simulations measure the consumed power during search operations by integrating current from all power supplies. As the evaluated IP routing application handles digital signals, digital-to-analog (DAC) converters are included in the analysis. The DACs impose additional overhead in terms of both chip area and energy, but the analysis below shows that the DAC overhead is not overwhelming. Importantly, for applications handling analog signals directly, digital-analog signal conversion is not required at all with our analog CAM, suggesting a promising application space. In contrast, analog-to-digital (ADC) converters would be required when using digital SRAM-based TCAMs, and we note ADCs are usually much more expensive (area/power) than DACs.

The DAC we use in our evaluation is a simple 4-bit current-steering design, the schematic of which is shown in SUPPLEMENTARY FIGURE 10. There are two major portions in the DAC design, with the first being the circuit mirrors input for $I_{REF} \times 1$ and $I_{REF} \times 4$ respectively (shown in panel a). The input current mirrors are shared globally across all the DAC channels, while the current mirror outputs are selectively turned on to convert the digital inputs that are applied on the switch transistors to analog outputs (panel b). The simulated curves for the DAC operation are shown in SUPPLEMENTARY FIGURE 10c.”

While unoptimized, the total energy consumption during search operations in an 86×12 analog CAM array is estimated to be 0.52 fJ per search per cell according to the simulations (SUPPLEMENTARY TABLE 2), and will be smaller in practice due to larger arrays and their reduced average DAC peripheral cost. If we convert the performance number in the aforementioned search table for IP routing, the same search function in the 21×16 TCAM table will consume 12.48 fJ per search in a 6×4 analog array, leading to a 0.037 fJ per search per equivalent TCAM bit. Additional energy improvements are expected for applications that handle analog signals directly.”

Supplementary Figure 10: A DAC design example for analog CAM. The circuit schematic for the modified current-steering DAC design with a, shared input current mirrors for I_{REF} and $I_{REF} \times 4$ and b, the output current mirrors that converts the digital outputs to the analog output signal. c, The simulated DAC operation with different digital inputs.

The power consumption, in fact, is very difficult to fairly compare because the state-of-art SRAM based implementation includes many power saving techniques with optimizations based on particular application use cases that are frequently unpublished. Therefore, we simulated the energy consumption and gave a specific power consumption number for our worst-case scenario. A more detailed power consumption for a particular use case will be the focus of a future work.

Reviewer #2 (Remarks to the Author):

This work describes the concept of analog CAM and implements the proof-of-concept demonstration. The 6T2M analog CMA cell is proposed to realize the function, and the analog CAM cell takes advantages of multi-level memristor conductance to represent different matched ranges. In this manner, it stores multi-bits or a continuous range of content in CAM applications. The idea is interesting. However, the major results are shown by simulations which limits the impact of this work. More experimental data are needed to validate the application of analog CAM. Moreover, the benefits from the using of analog CAM and the comparison to conventional CAM should be carefully discussed. The detail comments are listed below:

RESPONSE: We would like to thank the reviewer for acknowledging that our idea is interesting and constructive comments that improved the manuscript. In the revised manuscript, we have added more experimental data, and made further comparisons with conventional CAMs assuming a more advanced technology node. Please kindly find details in the following point-by-point responses.

1. A potential important contribution of this manuscript is the experimental demonstration of the proposed 6T2M analog CAM cell. However, the exiting manuscript has some misleading description of the experimental part. Based on this reviewer's understanding, the authors only measured 1T1R devices. Did the authors physically constructed a 6T2M cell and measure the electrical data from the cell? If this is the case, the authors should clearly show how to integrate the experimental part and simulation part in the man text with figures. Also, this reviewer strongly suggests the authors should conduct the experiment with physical 6T2M cell.

RESPONSE: A fully integrated 6T2M cell was under design and not available at the time of the first draft. The reviewer is right that we measured 1T1R devices with test points between the transistor and the memristor, as was described in the original manuscript, to prove the concept works before a tape-out.

Following the reviewer's suggestion, we have now included the experimental measurement results from the physical 6T2M analog CAM cell in the revised manuscript, in Fig. 4 in the main text and Supplementary Figures 3&4.

2. As the author declared, novel memristor devices are suitable for analog in-memory-computing at low precision requirements. However, the application of analog CAM requires reliable and accurate interval states of memristor devices. However, the memristor device suffers from some non-ideal characteristics, such as device variances, conductance drift and so on. Please show more data about the reliability behavior of the proposed analog CAM. Besides, according to Fig. 4e, the experimental output exhibits significant distortion. Please explain and comment.

RESPONSE: We agree with the reviewer that the application of the analog CAM requires reliable and accurate interval states of memristor devices. As the reviewer mentions, some memristors suffer from conductance drift and/or device variations, which may reduce the benefits brought by memristors. Our devices, on the other hand, have exhibited very stable analog states after an iterative program-and-verify conductance programming. We have previously demonstrated 5-bit consecutive conductance states programming, and the states can

be sustained for more than 6 hours without any noticeable drift. More experimental data on how the memristor states affect the analog CAM performance is shown below in the Response to Comment 4.

Figure R7. F, Histogram of the initial difference between the target and measured conductance written into a 128×64 array. A fit of the peak to a normal distribution yielded a standard deviation of $6 \mu\text{S}$, with the peak maximum located at $-5 \mu\text{S}$. G, Room-temperature state retention and read disturb of the device states. The d.c. conductance states of all the devices were measured with a 0.2 V bias for 1,000 cycles, or a total of 6.4 h, showing no discernible drift in the plots. H, Histogram of the normalized standard deviation (s.d.), defined as the s.d. per conductance range ($100\text{--}900 \mu\text{S}$), for all measured states, which was fitted to a lognormal distribution. This shows that there are fluctuations during the read operation that can occasionally degrade the effective precision of an individual memristor, but 90% of the device states have a normalized s.d. less than 0.39%. Adapted from [1]

The iterative programming may require more time (and energy) to configure the search table in the analog CAM, but it is a one-time overhead. The dominant operations are in the search phase. In cases where programming speed is critical, we have previously demonstrated an optimized programming scheme by utilizing the series connected transistor as an on-chip low-parasitic current compliance for coarse control and fast programming pulses for fine control [2]. The coarse control has even been successfully used for in-situ training of multilayer neural networks, which is a programming-heavy application.

The experimental data in the original manuscript was based on a 1T1R device with a much larger transistor than our simulation. Therefore, the curve was different than what is shown in the simulation. The experimental data has now been replaced by the new data from physical 6T2M cells in a 180 nm tape-out.

[1] C. Li, *et al*, “Analogue signal and image processing with large memristor crossbars”, *Nature Electronics*, 1, 52, 2018

[2] E. J Merced-Grafals, *et al*, “Repeatable, accurate, and high-speed multi-level programming of memristor 1T1R arrays for power efficient analog computing applications”, *Nanotechnology*, 27, 365202, 2016

[3] C. Li, *et al*, “Efficient and self-adaptive in-situ learning in multilayer memristor neural networks”, *Nature Communications*, 9, 2385 (2018).

3. The performance comparison of analog CAM to conventional CAM is not fairly evaluated. When benchmarking the performance of the analog CAM, the cost of the necessary periphery blocks, e.g. the DACs, circuits to program and verify the cell conductance range, should be included and carefully discussed.

RESPONSE: We agree that a comprehensive comparison should include the peripheral circuitry if the aim of the work is to directly replace the digital counterpart. In the revised manuscript, we

have included a customized current-steering type 4-bit DAC when discussing the energy performance. It may be further optimized in the future works, but it has already shown quite significant improvement over the digital counterpart for IP routing application.

The following discussion has been added to the revised manuscript:

“To evaluate the search energy improvement with an analog CAM, we simulated the circuit current from all the power supplies with an 86×12 analog CAM array. The cumulative consumed energy is calculated by integrating the voltage and current over the 16-cycle search with all MLs discharging in the search for the worst-case scenario. Estimating the full array power (including drivers and unoptimized peripherals such as the digital-to-analog (DAC) converters), the average total energy per search is ~ 0.52 fJ per analog cell, or 0.037 fJ for the equivalent number of TCAM bits implementing the same function (see SUPPLEMENTARY SECTION 5 for details). The energy per cell consumption is significantly smaller than an SRAM TCAM (0.165 fJ)³⁶, which utilizes numerous power saving techniques that provides a $>10 \times$ reduction but are not implemented in our analog CAM yet, and a conventional memristor TCAM (0.17 fJ)³⁷.”

With details described in the Supplementary Information:

“To evaluate the search energy consumption, our analog CAM array simulations measure the consumed power during search operations by integrating current from all power supplies. As the evaluated IP routing application handles digital signals, digital-to-analog (DAC) converters are included in the analysis. The DACs impose additional overhead in terms of both chip area and energy, but the analysis below shows that the DAC overhead is not overwhelming. Importantly, for applications handling analog signals directly, digital-analog signal conversion is not required at all with our analog CAM, suggesting a promising application space. In contrast, analog-to-digital (ADC) converters would be required when using digital SRAM-based TCAMs, and we note ADCs are usually much more expensive (area/power) than DACs.

The DAC we use in our evaluation is a simple 4-bit current-steering design, the schematic of which is shown in SUPPLEMENTARY FIGURE 10. There are two major portions in the DAC design, with the first being the circuit mirrors input for $I_{REF} \times 1$ and $I_{REF} \times 4$ respectively (shown in panel a). The input current mirrors are shared globally across all the DAC channels, while the current mirror outputs are selectively turned on to convert the digital inputs that are applied on the switch transistors to analog outputs (panel b). The simulated curves for the DAC operation are shown in SUPPLEMENTARY FIGURE 10c.”

While unoptimized, the total energy consumption during search operations in an 86×12 analog CAM array is estimated to be 0.52 fJ per search per cell according to the simulations (SUPPLEMENTARY TABLE 2), and will be smaller in practice due to larger arrays and their reduced average DAC peripheral cost. If we convert the performance number in the aforementioned search table for IP routing, the same search function in the 21×16 TCAM table will consume 12.48 fJ per search in a 6×4 analog array, leading to a 0.037 fJ per search per equivalent TCAM bit. Additional energy improvements are expected for applications that handle analog signals directly.”

Supplementary Figure 10: A DAC design example for analog CAM. The circuit schematic for the modified current-steering DAC design with a, shared input current mirrors for I_{REF} and $I_{REF} \times 4$ and b, the output current mirrors that converts the digital outputs to the analog output signal. c, The simulated DAC operation with different digital inputs.

On the other hand, however, the analog CAM circuit we proposed here may be applicable to some analog computing scenarios where analog signals are directly used as inputs, for example, an analog decision tree application proposed in the revised text on Page 18-19, 45-46 (see following shaded paragraph for the reviewer's convenience) or applications which may provide raw data from sensors as the input to the analog CAM. If a digital CAM was used for these kinds of applications, the input analog signals need to be converted to digital, which imposes additional energy and speed overhead. In addition, the non-volatility of memristor entirely removes the need of static power to maintain the storage of the matching table. Therefore, the final energy and speed comparison will be fairly complicated depending on a particular application. In order to make an apples-to-apples comparison, we only compare the dynamic power in the core-part of the CAM operation, which shows orders of magnitude benefits.

On Page 18-19:

“In addition to serving as a higher data-density digital replacement, the proposed analog CAM offers novel applications when the range search capability is utilized. As an example, a decision tree with binary and non-binary classification features can be implemented in the analog CAM directly by mapping each root to leaf path to a row in the analog CAM (see SUPPLEMENTARY FIGURE 11 for mapping details). Logically, each root-to-leaf path traverses a series of nodes with Boolean ANDs between elements in a given input feature vector (SUPPLEMENTARY FIGURE 11a). Since AND is commutative, we can reorder the nodes such that feature variables are processed in the same order for all paths³⁸. Nodes for the same feature are combined into one node and “don’t care” nodes can be inserted for features absent from a specific path, such that each path is of equal length. This representation can then be directly mapped to the analog CAM array, with each root to leaf path a row (see SUPPLEMENTARY FIGURE 11). As the matching row can directly drive the readout of the classification result, tree traversal becomes a one-cycle operation (SUPPLEMENTARY FIGURE 11b). This high throughput and low latency operation are highly advantageous and differentiated from current usage. While ensemble tree-based models are a popular state-of-the-art machine learning approach for classification and regression across diverse real-world applications, these models are difficult to optimize for fast runtime without accuracy loss in standard architectures³² due to non-uniform memory access patterns, resulting in unpredictable traversal and classification times today. With our proposed analog CAM, it becomes feasible to process large tree-based models at high data rates, such as those required for streaming applications or autonomous vehicles.”

On Page 45-46:

Decision trees with binary and non-binary classification features can be implemented in the analog CAM directly by mapping each root to leaf path to a row in the analog CAM. Logically, each root-to-leaf path traverses a series of nodes with Boolean ANDs between elements in a given input feature vector (SUPPLEMENTARY FIGURE 11a). Since AND is commutative, we can reorder the nodes such that feature variables are processed in the same order for all paths³⁸. Nodes for the same feature are combined into one node and “don’t care” nodes can be inserted for features absent from a specific path, such that each path is of equal length. This representation can then be directly mapped to the analog CAM array, with each root to leaf path a row (see SUPPLEMENTARY FIGURE 11). As the matching row can directly drive the readout of the classification result, tree traversal becomes a one-cycle operation (SUPPLEMENTARY FIGURE 11b). As each of the split outcomes of the tree are mutually exclusive, only one root-to-leaf path, or one row in the analog CAM, will “match” for a given feature vector. A collection of analog CAM arrays for decision trees with a local direct classification lookup can be used to implement ensemble tree-based models which are popular machine learning models.

Figure R8. Decision Tree in an analog CAM and a memristor RAM for fast tree traversal. Taken from the revised Supplementary Figure 11

We very much agree that the programming of the memristors may be expensive, in terms of both chip area and speed/energy. However, these are a one-time overhead, and programming circuits can be shared among many devices/arrays. One of the main points of our application is that the memristors do not need to be frequently updated, and therefore this overhead is not the majority. For scenarios requiring frequent programming, more targeted research and design optimizations would be required, but is beyond the scope of the present work.

The following paragraph has been added in the revised manuscript on Page 29.

“Iterative programming of memristors may take a long time to complete and consume significant amount energy. However, the applications proposed in this work do not require frequent updates of memristor conductances and therefore the overhead of programming memristors is negligible. In addition, due to the nonvolatility of memristor devices, the analog CAM does not require frequent re-programming or stand-by power once programmed. In the case of applications that do require frequent updates (e.g. in situ training of a decision tree), the peripheral circuit design would need to be carefully designed and optimized.”

4. The manuscript takes non-ideal factors of transistors into account, such as sub-threshold leakage current, etc., but memristors also have non-ideal factors, such as device read noise, retention, etc. The authors should discuss those non-ideal factors impact on the analog CAM search results, in other words, on the upper and lower bounds.

RESPONSE: Following the reviewer’s suggestion, we have conducted experimental measurements on the newly integrated analog CAM chip. From the experiment we learned that the original designed V_{hi} voltage of 0.8 V might be too high for our nanoscale memristor, which can cause read disturbance as the reviewer suggested. By reducing V_{hi} to 0.5 V during the search operation, we experimentally confirmed the memristor stays undisturbed throughout the experiments, while also helping to reduce the energy consumption.

In the revised manuscript, we added new experimental data about the read noise and retention. After programming a range into one analog CAM cell, we read the match line discharging current with different data line voltages (input data patterns), and the result shows that the lower and upper bounds did not change during 1,000 searches over more than 8,000 seconds. We have added the new results in the new text.

Figure R9. (left) The retention / reliability test shows that the cell maintains the searching range for more than 8,000 seconds with 1,000 measurement. The extracted searching range is shown in FIGURE 4f. (right), The range was extracted from the pull-down current, where a high pull-down current indicates a mismatch. The range stayed unchanged for 1,000 searched over more than 8,000 seconds. More data is shown in SUPPLEMENTARY FIGURE 5

5. Similar to comment 1, how will the practical imperfections of the volatile memristors affect the system performance in the 4T2M2S analog CAM model, Figure 6? The simulation is the ideal case and the authors should include the discussion of non-ideal factors. Meanwhile, is it possible to realize an inverter with 0.4V logic output under 180nm rule (i.e. the Mlso)? Please explain.

RESPONSE: The imperfection of volatile memristors, in particular the limited endurance performance will affect the system performance. It is possible that the diffusive (Ag-based) memristor may age during the search operation (the cited reference still showed $>10^8$ endurance cycles, see below), but devices with other mechanism, such as tunneling, or IMT, are not likely to suffer the same problem. The goal of this part of the work is to provide a possible improvement direction for future studies, and at the same time give a motivation to develop a device with a specific property. To avoid confusion, we have moved this to the supplementary information in the revised manuscript.

Figure R10, (b) Endurance of the selector with over 108 cycles. The inset shows the waveform employed in the endurance measurement which consists of a 10 μ s pulse with an amplitude of 4 V for ON switching followed by a 0.1 V read pulse. The time interval between switching pulses is 100 μ s (25 μ s waiting time plus 75 μ s read time). A series resistor of 3.9 M Ω was used to limit the ON-state current. Taken from [1]

We implemented a 0.8 V logic (buffers >0.4 V signals to logic 1, and <0.4 to logic 0) under 180 nm design rule with a specially doped transistor with lower threshold voltage. The size of the transistor is therefore larger than one with regular threshold voltage. The revised manuscript also included 16 nm designs, which uses a VDD close to 0.9 V, and therefore it is easy to implement 0.8 V logic. We acknowledge these helpful comments from the reviewer.

[1] R. Midya, et al. “Anatomy of Ag/Hafnia-based selectors with 1010 nonlinearity”. *Advanced Materials*, 29, 1604457 (2017).

6. Could the author explain more about how the non-linear effects are introduced when deducing the analytic expression of VDL in page 12?

RESPONSE: The analytic expression did not take into consideration the non-linear effects, which becomes quite complicated in the current-voltage (IV) relation for both memristor and transistor when non-linear since we cannot use a simple voltage dividing equation. Therefore, in practice, we use numerical simulations to determine the memristor conductance to be programmed. We revised the corresponding description in the main text to avoid confusion on Page 11.

“This analysis is consistent with the results shown in FIGURE 3d when the transistor is working in the linear regime, but when the memristor conductance is very small (i.e. the transistor voltage drop is small) a numerical simulation is required to extract the precise relation. In practice, we use the simulated relation in FIGURE 3d to determine the programming target memristor conductance values from the desired stored analog value or range.”

7. This article introduces the circuit design of analog CAM based on non-volatile memory, and compares its advantages with the traditional digital CAM. However, there has been analog CAM design before. What is the advantage of this article over previous design (Analog content addressable memory (CAM) employing Analog nonvolatile storage, US patent 6985372, 2006)?

RESPONSE: Thanks for pointing out the previous patent that discloses an analog content addressable memory design. The work introduces a concept to compare the analog storage in an EEPROM cell and the input voltage with an active comparator in each cell. In this design, the input voltage needs to precisely match the analog storage in the EEPROM cell and therefore it is quite challenging to be implemented in practice. It is also not clear how this design forms an

array for a practical application, nor would the performance (area, energy, speed) be expected to be competitive.

Our design, on the other hand, does not involve a comparator in each cell, and therefore is significantly more compact. Instead of storing discrete analog values, our design stores ranges encoded in two analog storage cells (memristors), providing more robust operation and allowing fuzziness. Following the reviewer's suggestion, we have added the following sentence (colored in blue) on Page 4 in the revised manuscript to acknowledge the previous pioneer work.

“However, nearly all memristor-based CAM designs utilize schemes similar to conventional static random-access-memory (SRAM) designs where the memristor only encodes binary states. The highly tunable analog conductance in memristor devices, with many stable intermediate states are not leveraged²⁹. An analog CAM design was proposed more than a decade ago that matches an input voltage with precise values stored in analog storage cells³⁰, but has not been implemented likely due to practical concerns of high power and area as it requires large numbers of active comparators and inefficient array implementations.”

Reviewer #3 (Remarks to the Author):

This paper proposes the nonvolatile analog TCAM with 6T2M or 4T4M schemes. The idea is novel and interesting; however, more results are needed to demonstrate the novelty and potential of this work. Therefore, the authors should revise their manuscript to address the concerns below before a final decision is reached.

1. The authors should show 6T2M/4T4M cell with feature size as the unit.

RESPONSE: The original manuscript uses the number of transistors as a reference to compare the occupied chip area, because we did not use the minimum sized transistors, in order to overprovision for the potential of high programming currents. It was a very conservative design that placed a priority on flexibility. The goal of this work is to explore the new concept and the corresponding device requirements.

In the revised manuscript, we have added the measurement data from a new tape-out at 180 nm technology node with integrated nanoscale memristors. The results show that the memristors can be reliably programmed with $\sim 100 \mu\text{A}$ current (as shown in the newly added Fig. 4c), therefore the oversized transistors are not necessary for future designs. We have also designed a new cell under TSMC 16 nm technology node, with 2 fins in all the n-type MOS FinFETs. As a result, the new design occupies $0.52 \mu\text{m}^2$ per analog cell, which is around $2,025 F^2$ if we define 16 nm as the feature size under the technology node. This is in contrast with $0.70 \mu\text{m}^2$ or $2,734 F^2$ (if we define 16 nm as the minimum feature size under this technology node, although the physical minimum feature size is in fact larger than 16 nm) for a reference SRAM cell that stores ternary information. However, we note that this feature size definition does not capture that fact that our cell stores analog information, as compared to the digital information stored in the SRAM cell noted. This discussion has been incorporated in the following analysis in the revised manuscript on Page 17.

As an area improvement, we estimate an $18.8\times$ reduction for our analog CAM $12.5 \mu\text{m}^2$ table compared to an SRAM implementation $235.2 \mu\text{m}^2$ (see SUPPLEMENTARY SECTION 5 for details).

On Page 41-42

Assuming a TCAM implementation with $0.70 \mu\text{m}^2$ per TCAM cell area overhead in a standard library under the same 16 nm technology node, the proposed range search functionality occupies $235.20 \mu\text{m}^2$ chip area while only $12.48 \mu\text{m}^2$ with our analog CAM, leading to $18.8\times$ reduction in area. The decreased area reduction comparing to that in transistor count is due to the fact that the reference SRAM layout utilizes the foundry's SRAM-specific layout rules, while our present analog CAM layout follows a more conservative logic rule.

2. The authors give a clear view of the concepts and explanation of the aTCAM cell and implementation. However, the authors should use the cell to build more applications in addition to IP routing to differentiate this work from the existing nonvolatile TCAM, because the latter also improves the density. Perhaps, as the authors propose, the fuzzy logic or probabilistic processing.

RESPONSE: We would like to thank the reviewer for the suggestion. In the revised manuscript, we have added the following paragraph to propose an in-memory computation approach for decision tree classification with our analog CAM.

On Page 18-19:

“In addition to serving as a higher data-density digital replacement, the proposed analog CAM offers novel applications when the range search capability is utilized. As an example, a decision tree with binary and non-binary classification features can be implemented in the analog CAM directly by mapping each root to leaf path to a row in the analog CAM (see SUPPLEMENTARY FIGURE 11 for mapping details). Logically, each root-to-leaf path traverses a series of nodes with Boolean ANDs between elements in a given input feature vector (SUPPLEMENTARY FIGURE 11a). Since AND is commutative, we can reorder the nodes such that feature variables are processed in the same order for all paths³⁸. Nodes for the same feature are combined into one node and “don’t care” nodes can be inserted for features absent from a specific path, such that each path is of equal length. This representation can then be directly mapped to the analog CAM array, with each root to leaf path a row (see SUPPLEMENTARY FIGURE 11). As the matching row can directly drive the readout of the classification result, tree traversal becomes a one-cycle operation (SUPPLEMENTARY FIGURE 11b). This high throughput and low latency operation are highly advantageous and differentiated from current usage. While ensemble tree-based models are a popular state-of-the-art machine learning approach for classification and regression across diverse real-world applications, these models are difficult to optimize for fast runtime without accuracy loss in standard architectures³² due to non-uniform memory access patterns, resulting in unpredictable traversal and classification times today. With our proposed analog CAM, it becomes feasible to process large tree-based models at high data rates, such as those required for streaming applications or autonomous vehicles.”

On Page 45-46:

Decision trees with binary and non-binary classification features can be implemented in the analog CAM directly by mapping each root to leaf path to a row in the analog CAM. Logically, each root-to-leaf path traverses a series of nodes with Boolean ANDs between elements in a given input feature vector (SUPPLEMENTARY FIGURE 11a). Since AND is commutative, we can reorder the nodes such that feature variables are processed in the same order for all paths³⁸. Nodes for the same feature are combined into one node and “don’t care” nodes can be inserted for features absent from a specific path, such that each path is of equal length. This representation can then be directly mapped to the analog CAM array, with each root to leaf path a row (see SUPPLEMENTARY FIGURE 11). As the matching row can directly drive the readout of the classification result, tree traversal becomes a one-cycle operation (SUPPLEMENTARY FIGURE 11b). As each of the split outcomes of the tree are mutually exclusive, only one root-to-leaf path, or one row in the analog CAM, will “match” for a given feature vector. A collection of analog CAM arrays for decision trees with a local direct classification lookup can be used to implement ensemble tree-based models which are popular machine learning models.

Figure R11. Decision Tree in an analog CAM and a memristor RAM for fast tree traversal. Taken from the revised Supplementary Figure 11

3. For TCAM, nanosecond order of magnitude is too slow. Even this paper it is proof of the concept, the authors better show the potential way to boost the search speed.

RESPONSE: The search latency is dominated by the match line capacitance and the discharging current in the case of one-cell mismatch. As discussed earlier, the original design is optimized for flexibility instead of performance, therefore the over-sized transistor results in a large match line capacitance and a tens of nanoseconds latency. However, we agree with the reviewer that a proof of concept work should also show the future scalability potential of performance when a more advanced technology node is available. The speed can be optimized by either reducing the match line capacitance or increasing the mismatch pull-down current by increasing search voltages. In the revised manuscript, we have added simulation data based on a layout following TSMC's 16 nm design rule, and the result shows a one-bit mismatch latency of <50 ps in a 86×12 analog CAM array (see new Fig 3b and Supplementary Fig. 2).

Figure R12: a, A simplified schematic of one analog CAM cell with pre-charging p-type MOSFET attached to its ML. b, The timing diagram for a search operation, where ML pre-charging is initiated by setting PC high, and the search operation by SL hi. c, Simulated transient plot of the precharging and search operation in a 86×12 analog CAM array for two cycles. The ML is pulled down within 100 ps when the DL voltage mismatches the stored range and is kept high in the case of a match. [Taken from the newly added Supplementary Figure 2]

4. The authors better add a diagram or cross-section microscopic picture to demonstrate the monolithic integration with silicon transistors.

RESPONSE: The revised manuscript now includes the cross-section diagram as well as a microscopic image of the integrated 180 nm cell from the recent tape-out in the new Figure 4a and Figure 4b respectively. We thank the reviewer for the constructive comment.

Figure R13: Experimental demonstration. a, Cross-sectional diagram showing monolithic integration of memristor on top of CMOS circuits. b, An optical microscope image of an integrated analog CAM array with an on-chip precharging peripheral circuit.

Reviewers' comments:

Reviewer #1 (Remarks to the Author):

The authors have performed a major overhaul of the paper. Although the concept has remained the same, the paper has now compelling experimental data, and much more interesting simulations. The simulations are also better controlled. I recommend publication.

Minor comments:

I recommend finding another way of plotting the 3-D graphs (5c and S5c): this is not readable in 3-D. Overall I recommend enhancing the readability of the figures, e.g. by using larger fonts.

I still think that the SI about volatile devices should mention that the search operation will damage the devices. There is a lot of confusion in the community about this, so this needs to be clear.

The new DAC discussion in SI is interesting and this issue should be mentioned in body text too.

Reviewer #2 (Remarks to the Author):

It is satisfactory to see the author provide the experimental test data on the 6T2M analog CAM cell and the simulation results with 16nm technology node. The added simulation for decision tree demonstration is interesting. However, I am still concerned about the reliability performance for analog memristors in CAM applications. Currently, the mainstream CAM researches are based on stable binary memristor rather than the analog memristor, which is accounted to the non-ideal device characteristics in analog devices. How device variations, conductance drifts, yield issue, read fluctuations and read disturbance in analog memristors deteriorate the analog CAM performance is still not revealed. The author should statistically evaluate the device performance and investigate the effects on CAM performance.

(1) The raw data in Fig. R7 is based on array test with HfO₂ memristors, not the TaO₂ memristor. The switching range is different, i.e. 100uS-900uS for HfO₂ memristor and 400ns-200uS for TaO₂ memristor. The author should statistically characterize the device performance and evaluate the

influence on the analog CAM application. Even with HfO₂ memristors, the Fig. R7c clearly shows the appearance of some “bad” devices. Would these devices affect the CAM functions? How to deal with this?

(2) The author shows good retention performance over 8,000 seconds by single device test. Is the 8,000s enough for CAM applications? How about the retention performance with other conductance states? Please show more data in a longer time regarding more states and more analog CAM cells, and analyze the limitations.

(3) According to Fig. 4d, the threshold voltage for SET process ranges from 0.6V to 0.8V. Setting the SL_{hi} as 0.5V still seems high to disturb the memristor state. Please comment on this.

(4) The analog CAM design pose challenges for general technology node due to requirements for 1V or even lower logic supply. Please note this limitation in the text.

(5) The author states that “The integrated memristors have a wide $\sim 10^3$ range of conductance tenability (FIGURE 4c) and a programming voltage <1 V under direct-current (DC) sweeps (FIGURE4d), enabling us to validate the search operations experimentally”. I wonder whether these behaviors are captured through single device measurement or array test. How is the variability and how the device variability affects the analog CAM performance?

(6) Why the high voltages on ML are different between Fig. 3b (0.6 V) and Fig. 3c (0.8 V) with a same 0.4V VDL? Please comment.

Reviewer #3 (Remarks to the Author):

The authors have answered all my questions. I do not have more issues with the manuscript.

In the following point-by-point responses, the original reviewers' comments are in black fonts and our responses are in blue fonts. Quotes from the manuscript are shaded and the changes in the revision is highlighted with blue font.

Reviewer #1 (Remarks to the Author):

The authors have performed a major overhaul of the paper. Although the concept has remained the same, the paper has now compelling experimental data, and much more interesting simulations. The simulations are also better controlled. I recommend publication.

RESPONSE: We thank the reviewer for the positive comments.

Minor comments:

1. I recommend finding another way of plotting the 3-D graphs (5c and S5c): this is not readable in 3-D. Overall I recommend enhancing the readability of the figures, e.g. by using larger fonts.

RESPONSE: Thank the reviewer for this constructive comment. For the data shown in Fig. S5c, we have plotted the key information, which is the searching boundary voltages, into a 2D plot in Fig. 4f, and highlighted it in the figure caption. In addition, we replotted the existing figures in 3-D graphs (Fig. 5c and S5c) with different linewidth, color and increased the text size in the labels to increase the readability.

In the figure caption on Page 14:

“f, The analog CAM cell stored range was extracted from the pull-down current in SUPPLEMENTARY FIGURE 5 (and one trace in (e)) where a high pull-down current ($>5 \mu\text{A}$) indicates a mismatch. The range stayed unchanged for 1,000 searches over more than 8,000 seconds.”

We have also carefully checked and enlarged the text labels in the figures to improve the readability.

2. I still think that the SI about volatile devices should mention that the search operation will damage the devices. There is a lot of confusion in the community about this, so this needs to be clear.

RESPONSE: Agreed, we have added the following in the SI on Page 39.

“Although TS memristors in the literature may suffer from endurance limits, the following analysis is aimed at providing a direction for even further performance improvements, with future work needed to fully explore these trade-offs.”

3. The new DAC discussion in SI is interesting and this issue should be mentioned in body text too.

RESPONSE: This is a good point to highlight our efforts in DAC design. The following discussion has been added in the revised manuscript on Page 17-18.

“To evaluate the search energy improvement with an analog CAM, we simulated the circuit current from all the power supplies with an 86×12 analog CAM array. For a practical evaluation of all digital applications, we custom designed a digital-to-analog (DAC) converter, which imposes additional overhead in both chip area and energy, but our analysis in SUPPLEMENTARY SECTION 5 and SUPPLEMENTARY FIGURE 12 shows this overhead is limited to approximately 10%. The cumulative consumed energy is calculated by integrating the voltage and current over the 16-cycle search with all MLs discharging in the search for the worst-case scenario. Estimating the full array power

(including drivers and unoptimized peripherals such as the custom-designed digital-to analog (DAC) converters), the average total energy per search is ~ 0.52 fJ per analog cell, or 0.037 fJ for the equivalent number of TCAM bits implementing the same function (see SUPPLEMENTARY SECTION 5 for details).”

Reviewer #2 (Remarks to the Author):

It is satisfactory to see the author provide the experimental test data on the 6T2M analog CAM cell and the simulation results with 16nm technology node. The added simulation for decision tree demonstration is interesting. However, I am still concerned about the reliability performance for analog memristors in CAM applications. Currently, the mainstream CAM researches are based on stable binary memristor rather than the analog memristor, which is accounted to the non-ideal device characteristics in analog devices. How device variations, conductance drifts, yield issue, read fluctuations and read disturbance in analog memristors deteriorate the analog CAM performance is still not revealed. The author should statistically evaluate the device performance and investigate the effects on CAM performance.

RESPONSE: We thank the reviewer for the positive comment on the newly added content in our last revision. Following the reviewer’s suggestion, we conducted more experiments to collect the reliability performance of our 50 nm TaOx memristor, as detailed in the following point-by-point responses.

Nevertheless, we would like to point out that the reason why mainstream CAM researchers focus on binary memristor, in our view, is that there has not previously existed a compelling circuit for an **analog** CAM, which is the main message of this work. On the other hand, while the analog memristor is indeed less mature than the digital counterpart at the moment, researchers are still heavily investing in applications based on accelerating matrix multiplication with analog memristors.

(1) The raw data in Fig. R7 is based on array test with HfO2 memristors, not the TaO2 memristor. The switching range is different, i.e. 100uS-900uS for HfO2 memristor and 400ns-200uS for TaO2 memristor. The author should statistically characterize the device performance and evaluate the influence on the analog CAM application. Even with HfO2 memristors, the Fig. R7c clearly shows the appearance of some “bad” devices. Would these devices affect the CAM functions? How to deal with this?

RESPONSE: We highlighted data from HfO2 memristors to demonstrate that devices with good analog performance already exist. Previously, we have also studied reliability performance of the same nanometer scale TaOx device [1] for analog conductance levels, and the data is shown in Figure R1. The TaOx memristor was switched to a conductance range from 0.1 nS to 600 μ S, and the read variation was smaller than 2 μ S.

In addition, following the reviewer’s suggestion, we have also measured the device statistics from a 64 \times 64 1T1R array on the same die where the analog CAM array was located. The array was programmed to a conductance range between 100 nS and 200 μ S, and the measured conductance stability distribution from 10,000 reads of all the devices is shown in Figure R2 (**also added as Supplementary Figure 6 in the revised manuscript**), from which one sees that the data agrees with our study on single device measurement that the read standard deviation (a measure for the device analog states’ stability), from the majority of devices/states, are within 2 μ S [1], and is better than that in our HfO₂ memristors in absolute standard deviation. [2].

Figure R1. Read instabilities studied at different conductance levels. a) Conductance as a function Read number at each state, showing conductance fluctuation, retention, and drifting trend. The magenta plot may involve an additional noise due to auto-gaining of the current measurement unit. b) Probability histogram for each programmed state. c) Standard deviation of conductance as a function of mean conductance. d) Coefficient of variation as a function of mean conductance. The conductance at all states are measured at $V_m = -0.11$ V and $V_g = 3.3$ V. Data is from a 25 nm device. (Taken from [1])

Figure R2. Distribution of the memristor device read stability shows that the measured (read) conductance for the majority of the memristor devices has a standard deviation of several μ S or smaller. The data was generated from conductance reads with 0.2 V read voltage from all devices in a 64×64 array for 10,000 times. The lateral size of the memristor is $50 \text{ nm} \times 50 \text{ nm}$. [Added as Supplementary Figure 6 in the revised manuscript]

Since the searching voltage bound is roughly proportional to the memristor conductance bound, as stated in Figure 3 and Equation 1 in the manuscript, a conductance change of 2μ S leads to about 1% change in the programmed acceptable searching range, limiting the average searching bit accuracy to 5-6 bit, which

is much higher than what a 3-4 bit analog CAM requires. As the reviewer has pointed out, there are significant outliers that exhibit a standard deviation around 8 μS , but it is still sufficient to differentiate 12 discrete states, i.e. 3-4 bit. In the case that significant outliers are not tolerable, we could disable the entire row by marking that the corresponding row is ‘bad’, similar to how a mainstream commercial memory functions. For some applications operating completely in the analog domain, for example, our proposed decision tree with analog CAM, a certain degree of defect tolerance to the outlier devices is also inherent. A comprehensive evaluation of these issues is the focus of future work.

[1] Xia, S. *et al*, Low-Conductance and Multilevel CMOS-Integrated Nanoscale Oxide Memristors, *Advanced Electronic Materials*, 5, 1800876 (2019)

(2) The author shows good retention performance over 8,000 seconds by single device test. Is the 8,000s enough for CAM applications? How about the retention performance with other conductance states? Please show more data in a longer time regarding more states and more analog CAM cells, and analyze the limitations.

RESPONSE: Whether 8,000 seconds is good enough depends on the specific application scenario, because the devices can be ‘refreshed’ by infrequent programming at a reasonable cost. We agree with the reviewer that retention performance is critical for a memristor device to be used in analog CAM and other forms of analog computing. Following the reviewer’s suggestion, we further measured the retention data after programming 380 devices into a variety of conductances between 0-200 μS for more than 20 hours. The data is shown in Figure R3 (and the new Supplementary Figure 7 in the revised manuscript), which shows no systematic drift in conductance.

Figure R3. Multilevel retention performance of our integrated TaO_x device shows the device conductances did not drift for over 20 hours under room temperature. Each datapoint is averaged from 50 repeated reads. [Added as Supplementary Figure 6 in the revised manuscript]

(3) According to Fig. 4d, the threshold voltage for SET process ranges from 0.6V to 0.8V. Setting the SL_{hi} as 0.5V still seems high to disturb the memristor state. Please comment on this.

RESPONSE: We hope we can clarify these issues as the conditions of Fig. 4d have important differences from those of the memristors in the aCAM circuit. Indeed, if a positive voltage is applied, ones sees from

4d that even a ~ 0.5 V voltage could disturb the memristor conductance state. However, the negative threshold (RESET) voltage has much less variability than the SET voltage, and small voltages usually do not disturb memristor states. This is part of the reason why during the search operation, the memristor in the analog CAM is negatively biased. Importantly, the IV sweep data is from a direct-current (d.c.) measurement, while in the analog CAM search operation, very rapid pulses are applied to the SL_hi with less disturbance.

In addition, since there is a transistor connected in series, the actual voltage drop across the memristor device is always, in fact, much smaller than 0.5 V. When the analog CAM is operated in the search mode, the gate voltage (V_{DL}) is always smaller than 1 V while 3.3/5 V is used during programming to fully turn ON the transistors. Fig. R4a shows how memristor conductance states get disturbed by applied voltage pulse with transistors fully turned ON (**also added as Supplementary Figure 7 in the revised manuscript**). The data is consistent with what we have shown in Fig. 4d, where the RESET voltage is very close to 0.5 V. On the hand, when the gate voltage of the series transistor is 1 V, which is the maximum V_{DL} value in the search mode, the memristor conductance states are not disturbed by the applied voltage pulses.

Figure R4. Device stability tests with different voltages on SL hi. The memristors are RESET by voltage (>0.5 V) pulses applied to SL hi with the series transistors fully turned ON. The initial value of the memristors was programmed between 100-200 μS . b, On the other hand, the memristor states were not changed by the applied pulses when in the search mode where the V_{DL} is always smaller than 1 V. [Added to as Supplementary Figure 7 in the revised manuscript]

To validate our statement, we further measured the device statistics after reading all device conductances in a 64×64 array. Figure R4 shows the distribution of the conductance change after 10,000 read operations on each device with the read voltage specified in the title of the corresponding panel, which shows only normal amount of read noise (with majority of devices smaller than 2 μS), instead of systematic drift towards one direction cause by large read voltage. **The data has also been added as Supplementary Figure 7 in the revised manuscript.** Finally, we note that the devices built here have been intentionally tuned to have lower switching voltages (Fig 4d), as disturb effects were not a priority for these demonstrations. Through thickness and doping engineering, it is possible to increase the switching voltages several hundred millivolts to reduce disturb, while maintaining ease of programming.

Figure R4. c-f, The device stability after read operations with different reading voltages. Each panel shows the distribution of the memristor conductance change after 10,000 repeated read operations with the read voltage specified in the title. The conductances do not show noticeable disturb by the read operations, within the noise of the read operation. [Added to as Supplementary Figure 7 in the revised manuscript]

(4) The analog CAM design pose challenges for general technology node due to requirements for 1V or even lower logic supply. Please note this limitation in the text.

RESPONSE: We would like to clarify that the specially doped transistor to enable lower voltage logic (mentioned in the previous round rebuttal letter) is only for implementing about 0.8 V logic under 180 nm technology node, while under 16 nm tech node, 1 V or a little lower than 1 V logic supply is standard. This low logic supply is only used in the sense amplifier circuit, which can be potentially replaced by a comparator with a low external voltage supply with minimum additional overhead.

In addition, we agree that given the memristor we optimized for this application requires an about 1 V programming voltage and slightly larger to form, therefore it poses challenge when future technology could not provide more 1 V or lower supply. We note this limitation in the revised text on Page 12:

“Given that the switching voltage of the memristor device exhibits a certain degree of variation, some devices may require a larger voltage to program than others (see SUPPLEMENTARY FIGURE 7a). Therefore, it may impose challenges for a future technology node which supplies a voltage smaller than 1 V.”

(5) The author states that “The integrated memristors have a wide $\sim 10^3$ range of conductance tenability (FIGURE 4c) and a programming voltage <1 V under direct-current (DC) sweeps (FIGURE 4d), enabling us to validate the search operations experimentally”. I wonder whether these behaviors are captured through single device measurement or array test. How is the variability and how the device variability affects the analog CAM performance?

RESPONSE: Fig. 4c and Fig. 4d are from single device measurements, but they are typical behavior among many devices we have measured. While we have performed more device study in the previous work [1], we would like to note the variability of conductance range and programming voltage is not very critical for read-dominated analog CAM operation, and Fig. 4c and 4d serves a purpose to explain how a typical device work to the general audience. Statistically, the data in Figure R3 shows that the devices in an array, can be programmed to a desired conductance pattern within the conductance range from 100 μ S to 200 μ S. We observe a certain degree of programming voltage variability, and it is also noticeable in Fig. 4d, as the switching voltages are not exactly the same between switching cycles. But it is not critical because we use iterative program-and-verify approach to configure the devices into target values, which does add programming/re-programming overhead. But owing to the non-volatile nature of the memristor device (shown in Figure R3), the memristors do not need to be frequently updated, therefore the additional overhead, we believe, is acceptable.

On the other hand, we do believe the device conductance read stability and retention performance is critical to the analog CAM operation, and we have provided an extensive study in Figs. R2, R3, R4, which have been added to the revised manuscript.

We have also revised the text to clarify the above points:

“The integrated memristors have a wide $\sim 10^3$ range of conductance tunability (FIGURE 4c) and a programming voltage < 1 V under direct-current (DC) sweeps (FIGURE 4d shows a typical switching curve), enabling us to validate the search operations experimentally. Given that the switching voltage of the memristor device exhibits a certain degree of variation, some devices may require a larger voltage to program than others (see SUPPLEMENTARY FIGURE 7a). Therefore, it may impose challenges for a future technology node which supplies a voltage smaller than 1 V.”

[1] Xia, S. *et al*, Low-Conductance and Multilevel CMOS-Integrated Nanoscale Oxide Memristors, *Advanced Electronic Materials*, 5, 1800876 (2019)

(6) Why the high voltages on ML are different between Fig. 3b (0.6 V) and Fig. 3c (0.8 V) with a same 0.4V VDL? Please comment.

RESPONSE: We acknowledge this was a mistake, where in Fig 3c, the y-axis is the buffered match line (ML) signal after the ML sense amplifier (MLso) (~ 0.88 V), same as those in Fig. 3e and Fig. 3f. Figure 3b shows the voltage on ML before the sense amplifier to show the transient discharging process. In the original version, we marked the y-axis in the panel c as voltage on ML instead of MLso by mistake, and it has been corrected in the revised version.

Reviewer #3 (Remarks to the Author):

The authors have answered all my questions. I do not have more issues with the manuscript.

RESPONSE: We appreciate the reviewer’s time in reviewing this manuscript.

REVIEWERS' COMMENTS:

Reviewer #2 (Remarks to the Author):

The authors have answered all my questions and concerns clearly. I would suggest to accept this manuscript

In the following point-by-point responses, the original reviewers' comments are in black fonts and our responses are in blue fonts.

Reviewer 2 (Remarks to the Author):

The authors have answered all my questions and concerns clearly. I would suggest to accept this manuscript.

RESPONSE: We thank the reviewer for the positive comments and appreciate the reviewer's time in improving our manuscript.